# Lack of Diaph3 relaxes the spindle checkpoint causing the loss of neural progenitors

Devid Damiani[1,2], André M. Goffinet[1,2], Arthur Alberts[3] & Fadel Tissir[1]

The diaphanous homologue Diaph3 (aka mDia2) is a major regulator of actin cytoskeleton. Loss of Diaph3 has been constantly associated with cytokinesis failure ascribed to impaired accumulation of actin in the cleavage furrow. Here we report that Diaph3 is required before cell fission, to ensure the accurate segregation of chromosomes. Inactivation of the *Diaph3* gene causes a massive loss of cortical progenitor cells, with subsequent depletion of intermediate progenitors and neurons, and results in microcephaly. In embryonic brain extracts, Diaph3 co-immunoprecipitates with BubR1, a key regulator of the spindle assembly checkpoint (SAC). Diaph3-deficient cortical progenitors have decreased levels of BubR1 and fail to properly activate the SAC. Hence, they bypass mitotic arrest and embark on anaphase in spite of incorrect chromosome segregation, generating aneuploidy. Our data identify Diaph3 as a major guard of cortical progenitors, unravel novel functions of Diaphanous formins and add insights into the pathobiology of microcephaly.

[1] Developmental Neurobiology Unit, Université catholique de Louvain, Institute of Neuroscience, Avenue Mounier 73, Box B1.73.16, Brussels 1200, Belgium. [2] Developmental Neurobiology Unit, WELBIO, Institute of Neuroscience, Avenue Mounier B1.73.16, Brussels 1200, Belgium. [3] Laboratory of Cell Structure and Signal Integration, Van Andel Research Institute, 333 Bostwick Avenue N.E., Grand Rapids, Michigan 49503, USA. Correspondence and requests for materials should be addressed to F.T. (email: fadel.tissir@uclouvain.be).

Formins constitute a family of 15 proteins in the mouse and human, characterized by the presence of two formin homology domains. By interacting with the growing ends of actin filaments, formins protect from capping, catalyse actin polymerization and regulate filament bundling into filopodia[1–4], supporting the establishment and maintenance of cell polarity during development and in response to disease. Diaphanous formins, known in mammals as Diaph1, 2 and 3 (Diaph1–3) are a subgroup of the formin family related to *Drosophila* diaphanous[5]. Diaph1–3 exist in two forms. In the inactive 'locked' form, the carboxy-terminal diaphanous autoregulatory domain interacts with the upstream inhibitory domain. Activation occurs through binding of a small GTPase to the GTPase binding domain, which disrupts the interaction between diaphanous inhibitory domain and diaphanous autoregulatory domain, and releases the protein ends[6].

In flies, mutations in the *diaphanous* gene generate defects in gametogenesis and neuroblast formation, with polyploidy attributed to compromised cytokinesis[7]. In mammals, *Diaph1–3* mutations have been associated with local actin cytoskeleton dysfunctions. For instance, in *Diaph1* and *2* double-knockout (*Diaph1–2 dko*) mice, whereas radial migration and layer formation of cortical excitatory neurons are largely unaffected, the tangential migration of cortical interneurons and neuroblasts from the adult neurogenic sub-ventricular zone (VZ) to the olfactory bulbs are impaired. F-actin organization is disrupted at the rear of migrating interneurons and neuroblasts, impairing the translocation of centrosome and nucleus[8]. Diaph3 accumulates in the cleavage furrow during transition from anaphase to telophase, and its depletion in dividing cells *in vitro* affects the amount of F-actin at the equatorial region[4,9]. In *Diaph3*-deficient mice, erythroid cells differentiate normally, but during the late divisions of erythroblasts, daughter cells fail to separate because of decreased accumulation of actin in the cleavage furrow[10]. Mice with overexpression of Diaph3 have anomalies in the inner, but not outer, hair cells. These mice exhibit hearing loss, suggesting an essential role for Diaph3 in assembly and/or maintenance of actin filaments in stereocilia[11]. In humans, a mutation in the region coding for 5′-untranslated region of the *DIAPH3* messenger RNA results in two to threefold overexpression of the protein, leading to delayed onset, progressive deafness known as auditory neuropathy non-syndromic autosomal dominant 1 (ref. 12). Furthermore, a double hit in the *DIAPH3* gene (a maternally inherited deletion on 13q and a point mutation in the paternal copy) was associated with autism[13]. In addition to its well-documented role in actin cytoskeleton, *in vitro* studies have implicated Diaph3 in the dynamics of microtubules (MTs). Diaph3 co-localizes with stable MTs and its overexpression is sufficient to generate and orient stable MTs[14]. Diaph3 can directly bind (and stabilize) MTs in an actin nucleation-independent manner[15,16]. Alternatively, by interacting with the MT tip proteins EB1 and adenomatous polyposis coli (APC), Diaph3 was proposed to serve as scaffold protein[17].

A key feature of the mammalian cortex is the substantial growth of its germinal zones. At early stages of cortical development, neuroepithelial (NE) cells proliferate rapidly by symmetrical division, to amplify the pool of progenitors[18]. A tight regulation of the cell division machinery is therefore required, to ensure a correct mitotic process and even segregation of chromosomes between daughter cells. Although intensive research in cortical development and evolution has identified numerous genes that influence cortical progenitor cell division, much effort is still needed to fully understand the underlying molecular mechanisms. Here we report that the formin Diaph3 acts early in mitosis to secure appropriate karyokinesis. Diaph3 belongs to a molecular network that comprises components of the spindle assembly checkpoint (SAC) and chromosomal passenger complex (CPC) machineries. This network regulates kinetochores–mitotic spindle interactions and controls the transition of cortical progenitors from metaphase to anaphase. Mutation of Diaph3 compromises the level of SAC activation. Hence, nuclear errors are not properly 'amended' by the spindle checkpoint, causing aneuploidy, cell death and cortical hypoplasia.

## Results

**Diaph3 ko mice display severe developmental defects.** We studied the expression of Diaph3 in the nervous system using *in situ* hybridization. The mRNA signal was diffuse at embryonic (E) day 10.5 and more confined to VZ of the cerebral cortex at E13.5 (Fig. 1a). Analysis using a fluorescent RNAscope probe showed that the signal was the highest in the outermost germinal zone, where radial glial and intermediate progenitor cells reside, and no signal was observed in doublecortin-positive neurons (Fig. 1b–e). To inactivate the *Diaph3* gene, we inserted a cassette containing FRT-En2-IRES-LacZ-loxP-neo-FRT-loxP in intron 9 (Supplementary Fig. 1a). Appropriately targeted embryonic stem cells were cloned and injected in *C57BL/6N* blastocysts. To validate the mutation, we used complementary DNA from wild-type (WT), heterozygous (*ko/+*) and *ko* mice as a template for reverse transcriptase–PCR (Supplementary Fig. 1b). A PCR amplifying exons 8–9 was used as a control (present in the cDNAs from the three genotypes). Two primer pairs spanning exons 9–11 and 13–15 were predicted to produce no amplicon in *ko* mice. The two pairs produced however amplicons even in the *ko*. Primers in exons 9 and 11 yielded a larger amplicon than in WT. Sequencing of this amplicon showed the presence of a cryptic splicing donor site located 115 nucleotides downstream of the beginning of exon En2. The mRNA was spliced to exon 10, as in WT mice, but included 115 nucleotides from the cassette, creating a frameshift mutation with 3 stop codons in exon 10 (Supplementary Fig. 1c). This suggests that no functional Diaph3 protein could be produced in the *ko* allele. To verify this, we analysed protein extracts from E13.5 embryos of WT, heterozygous and *ko* mice by western blotting using a rabbit polyclonal antibody against the C-terminal segment of the Diaph3 protein (Fig. 1f). The antibody disclosed a doublet in the 130 kDa range in WT and heterozygous, but not in *ko* tissue (predicted Diaph3 molecular weight: 134 kDa), confirming the absence of the C-terminal fragment of the protein. *Diaph3*[ko/+] heterozygous mice were viable and fertile. Very few *Diaph3 ko* mice were born (5 instead of ∼200 expected in 803 genotyped pups), indicating that the homozygous mutation is mostly embryonic lethal. Surviving animals displayed small brain and body size, as well as facial deformities (Fig. 1g–i). In addition, hydrocephalus was found in three out of the five mutants (Supplementary Fig. 2a).

As most of the Diaph3 mutants died before birth, we compared *ko* and control (WT or heterozygous) embryos at different developmental stages. They looked quite similar up to E10.5 (Fig. 2a). From E11.5 onwards, *ko* embryos consistently exhibited developmental abnormalities, with small size, bending of the body axis and a drastic atrophy of the lower jaw (Fig. 2b). Mutant embryos with the most severe phenotype were also characterized by an axial tortuosity of the neural tube, ranging from a moderate bending to a clear zig-zag pattern (Fig. 2c). Based on the expression profile and brain phenotype, we investigated the development of the cerebral cortex. At E13.5, there was a consistent reduction of the size of dorsal telencephalon (Fig. 2d). The phenotype varied from mild to severe expressivity, yet occurred with a 100% penetrance. Moreover, an increase

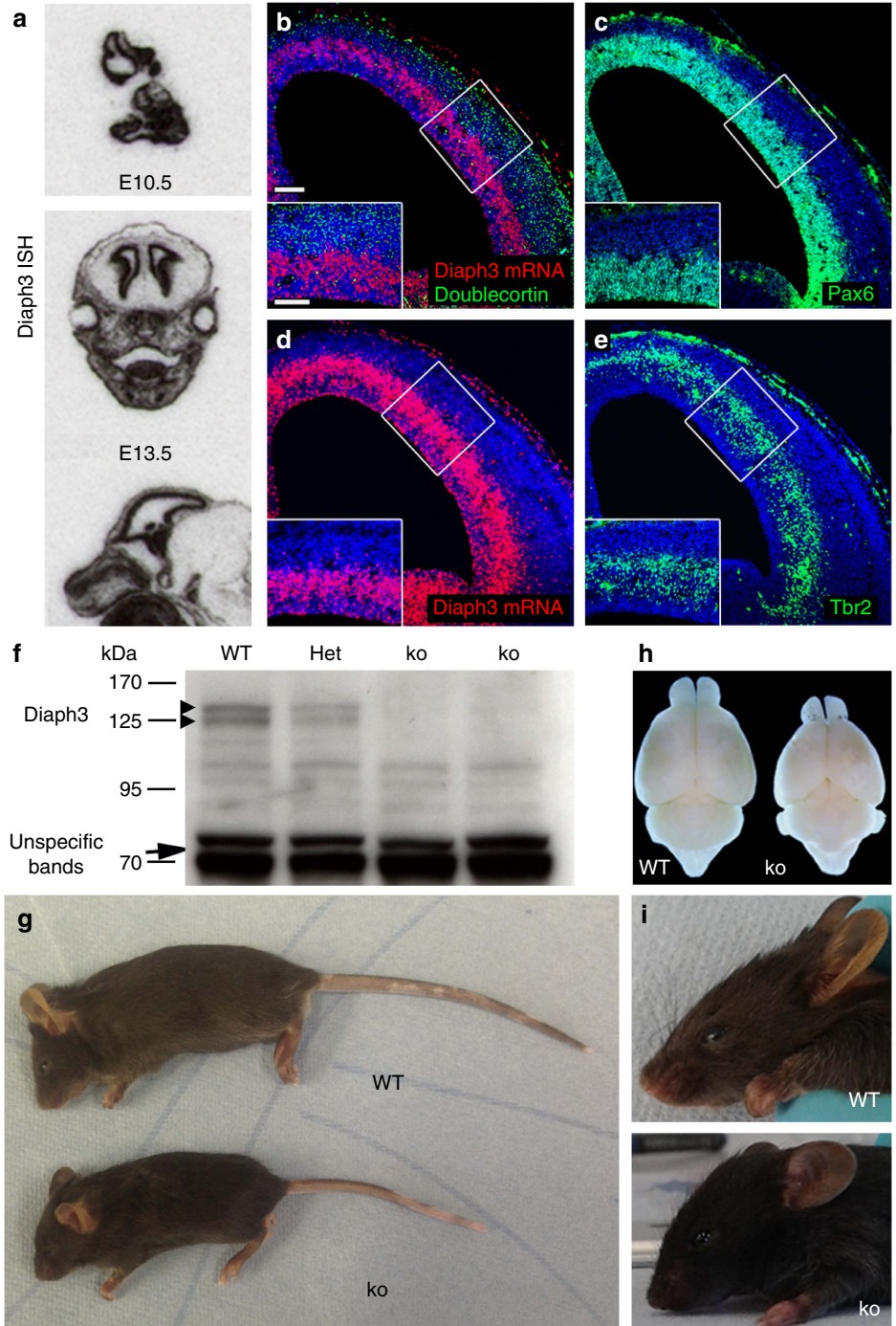

**Figure 1 | Generation of Diaph3 mutant mice.** (**a**) Sections of E10.5 embryos and E13.5 heads, hybridized with a *Diaph3* radiolabelled probe. The mRNA signal is associated with the telencephalic VZ where the progenitor cells reside. (**b–e**) Coronal sections of E13.5 brain hemispheres hybridized with a fast red-labelled RNAscope probe. The signal localized mostly in the outer part of the VZ (**b,d**). No signal was found in committed cortical neurons stained for Doublecortin (green, **b**). (**c,e**) Sections adjacent to those illustrated in **b,d**, stained for Pax6 (**c**) and Tbr2 (**e**). Scale bars, 100 µm and 50 µm in the insets. (**f**) Western blotting of E13.5 embryo lysates from WT, heterozygotes (Het) and mutant (*ko*) littermates. The absence of Diaph3 protein doublet in *Diaph3 ko* mice (arrowheads) is noteworthy. Nonspecific bands at 70 kDa provide a loading control (arrow). (**g–i**) Comparison between *Diaph3* WT and *ko* littermates at P25. The small body size (**g**), small brains (**h**) and facial deformities (**i**) in *Diaph3 ko* mice are noteworthy. ISH, *in situ* hybridization.

in both the number and size of blood vessels was observed in haematoxylin–eosin- and Isolectin B4-stained sections (Fig. 2e and Supplementary Fig. 2b). *Diaph3 ko* embryos appeared whitish (Fig. 2f) and histological examination disclosed a substantial depletion of erythrocyte progenitors in hepatic erythroblastic islands that resulted in severe anaemia, confirming previous findings using a different mutant allele[10].

The large majority ($\sim$ 97.5%) of mutant embryos died between E12.5 and E14.5.

**Cortical hypoplasia is due to apoptosis of neural progenitors.** We analysed the cytoarchitecture of cerebral cortex by immunohistochemistry. At E13.5, the number of the three main

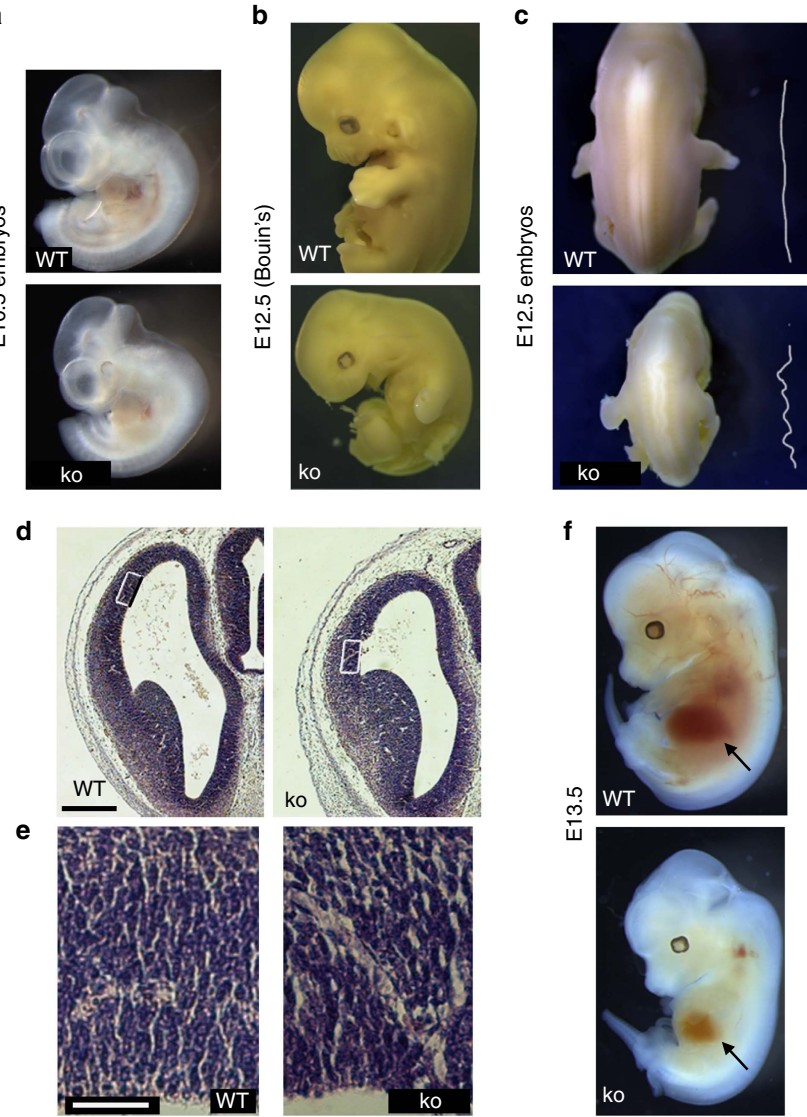

**Figure 2 | Developmental phenotype of *Diaph3* mutant mice. (a)** Development of *Diaph3* mutant mice was grossly normal until E10.5. **(b)** At E12.5, mutant embryos displayed a delayed development and facial malformation. **(c)** Neural tube deformity in *Diaph3* mutant embryos. A bending pattern was seen in severely affected mutants. **(d)** Haematoxylin–eosin-stained sections of E13.5 telencephalon from WT and *Diaph3 ko* littermates. The reduction of cortical size was already evident at this stage. **(e)** High magnification of the boxed areas in **d**, showing the enlargement of blood vessels in the mutant embryos. **(f)** Whitish appearance and impaired liver haematopoiesis in E13.5 *ko* compared with WT embryos. Scale bars, 500 μm (**d**) and 50 μm (**e**).

cell populations, namely apical (radial glial) progenitors (Pax6$^+$), intermediate progenitors (Tbr2$^+$) and neurons (Tbr1$^+$). The three populations were decreased in *Diaph3 ko* mice (Fig. 3a,b; quantification in Fig. 3c). Compared with WT littermates, the number of apical progenitors declined by ∼40% (number of Pax6$^+$ cells per 100 μm-wide stripe of frontal cortex: 243.75 ± 15.75 versus 377.40 ± 22; medial cortex: 188.6 ± 7.61 versus 299.44 ± 22.39; caudal cortex: 159.4 ± 4.81 versus 233.7 ± 20.31). A comparable reduction was observed in the number of intermediate progenitors (number of Tbr2$^+$ cells per 100 μm-wide stripe of frontal cortex: 140.32 ± 10.6 versus 238.66 ± 15.06; medial cortex: 103.74 ± 5.04 versus 182.3 ± 12.56; caudal cortex 103.84 ± 8.11 versus 145.05 ± 9.83) and postmitotic neurons (number of Tbr1$^+$ cells per 100 μm-wide stripe of frontal cortex: 100.78 ± 5.32 versus 157.24 ± 11.56; medial cortex: 71.95 ± 4.35 versus 125.7 ± 12.42; caudal cortex: 59.05 ± 6.34 versus 138.81 ± 11.53). Thus, the three cortical cell types were reduced in E13.5 *Diaph3 ko* embryos. To understand the developmental basis of cortical hypoplasia, we studied apoptosis

using immunohistochemistry for activated Caspase-3aCas and TdT-mediated dUTP nick end labelling (TUNEL). We observed several aCas3-positive cells in the mutant cortex at E13.5, whereas such cells were hardly seen in control littermates (Fig. 3d). We examined earlier stages and detected a massive cell death at E10.5 (Fig. 4a,b). Many cells were also positive for aCas9 (Fig. 4c), suggesting that mutant cells die via the mitochondrial apoptosis pathway[19–21]. Double immunostaining indicated that the dying cells (aCas3$^+$ or TUNEL$^+$) were neither neurons (Tuj1$^+$) nor intermediate progenitors (Tbr2$^+$), but rather Pax6$^+$ neural progenitors (Fig. 4d–g and Supplementary Fig. 3a,b).

Given the defective erythropoiesis in *Diaph3* mutant mice, we wondered whether the reduced erythrocyte concentration could trigger hypoxia and induce apoptosis of cortical progenitor cells. In E10.5 embryos, blood vessels have not yet invaded the neuroepithelium and remain confined to meninges. Brain tissue oxygenation depends on local diffusion rather than intra-parenchymal circulation. Nevertheless, the possibility that the oxygen pressure is perturbed in *Diaph3 ko* embryos cannot be

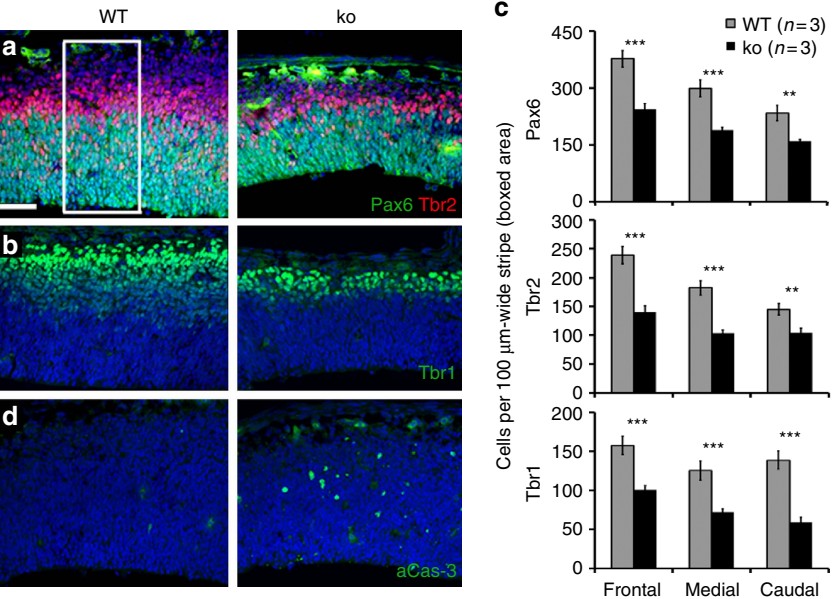

**Figure 3 | Different cell populations are affected in *Diaph3* mutant.** E13.5 cortical sections immunostained for Tbr2 and Pax6 (**a**), Tbr1 (**b**) and aCas3 (**d**). (**c**) Quantification of Pax6$^+$, Tbr2$^+$ and Tbr1$^+$ cells. Cells were counted in 317.13 µm-wide cortical areas from three mutants and three controls, and counts were expressed as cells per 100 µm-wide cortical stripe (boxed area in **a**). An average of 40% decrease in the three cell populations was seen in the mutant tissue. $n = 3$ ko and 3 WT embryos, Student's $t$-test, $^{**}P < 0.01$ and $^{***}P < 0.001$. Error bars represent s.e.m. Scale bar, 50 µm.

completely excluded. To test this, we injected pregnant females at E10.5, when cortical cell death peaks, with the hypoxyprobe-1 (pimonidazole hydrochloride), which binds covalently to proteins when the oxygen pressure is lower than 10 mm Hg. As a positive control, we used hypoxic tumours obtained by subcutaneous injection of human breast cancer-derived cells (B16F10) in mice. The staining with anti-pimonidazole antibodies was very low and comparable in *ko* and control cortices, whereas intense staining was visible in the tumour tissue (Supplementary Fig. 4). To definitely rule out a role of hypoxia-induced apoptosis, we generated cortex-specific mutants by crossing the *Diaph3* floxed allele (exons 10 and 11 are flanked by LoxP sites) with *Emx1-Cre* mice[22] (where the Cre recombinase is active in cortical cells from E9; Supplementary Fig. 1a). Whereas no apoptosis was seen in *Diaph3*$^{f/+}$;*Emx1-Cre* or *Diaph3*$^{f/f}$ cortices, *Diaph3*$^{f/f}$;*Emx1-Cre* embryos exhibited many apoptotic cells, demonstrating that cell death is caused by the lack of *Diaph3* in cortical progenitors rather than in blood cells (Supplementary Fig. 5a). The loss of cortical progenitors led to a significant reduction of cortical size at E15.5 and in the adult (Supplementary Fig. 5b–d).

**Mitotic errors generate aneuploidy and cell death.** As many apoptotic profiles scattered throughout the cortex resembled cell remnants with a round morphology and two or more micronuclei sporadically (Fig. 4b), we focused on mitotic cells located in the apical tier of the VZ. 4,6-Diamidino-2-phenylindole (DAPI) staining indicated the presence of asymmetric nuclear division (Fig. 4g). We examined 211 dividing (phospho-Vimentin$^+$) cells, among which 71 were apoptotic (aCas3$^+$ or TUNEL$^+$). Whereas only 6.4% (9/140) of non-apoptotic cells displayed mitotic errors, 98.5% (70/71) of apoptotic cells had such errors, strongly suggesting a correlation between apoptosis and chromosome segregation abnormalities in *Diaph3 ko* cortical progenitor cells (Fig. 4h and Supplementary Fig. 3c,d). These abnormalities produce mitotic catastrophe or give rise to aneuploid progenies[23]. To assess whether the dying cells, scattered in the *Diaph3 ko* dorsal telencephalon, were aneuploid offspring of aberrantly

dividing cortical progenitors, we quantified the genomic content of E11.5 telencephalic cells by DNA flow cytometry analysis. We found that, in addition to the 2C and 4C peak characteristics of euploid cells before and after DNA replication, respectively, a significant fraction of mutant cells exhibited a DNA content less than 2C (*ko*: $5.43 \pm 1.20\%$, $n = 21,233$ cells from 5 forebrains, control: $0.73 \pm 1.20\%$, $n = 125,123$ cells from 23 forebrains; Fig. 4i). Both asymmetric segregation and abnormal genomic content in the mutant tissue prompted us to analyse the karyotype of Diaph3-deficient neural stem cells, grown as neurospheres. We confirmed the neural identity by Nestin immunostaining (Supplementary Fig. 6). We counted the number of chromosomes in 129 control and 161 mutant metaphase spreads obtained from cells harvested at passage 8 (4 animals for each genotype). In control cells, 90% of metaphases (116/129) were euploid and the number of chromosomes in aneuploid metaphases comprised between 38 and 42 (Fig. 4j and Supplementary Table 1). In sharp contrast, the percentage of euploid metaphases decreased to 35% (57/161) in mutant cells. Among aneuploid cells, 16.1% (26/161) had < 38 chromosomes (Fig. 4k), 6.2% (10/161) had > 42 chromosomes (Fig. 4l) and 3.7% (6/161) were tetraploid (Fig. 4m). Taken together, these results suggest that the lack of Diaph3 disrupts nuclear segregation and triggers chromosomal instability, aneuploidy and loss of cortical progenitors.

**Absence of Diaph3 relaxes the spindle checkpoint.** During mitosis, segregation of chromosomes relies on the SAC machinery, which 'monitors' connections between spindle MTs and kinetochores, and delays the transition to anaphase until all chromosomes are correctly attached. Given the chromosomal instability in *Diaph3 ko* mice, we anticipated an overactivation of the SAC, with accumulation of dividing neural progenitors and overall increase in mitotic cell density. We tested this hypothesis using immunohistochemistry for phosphorylated histone H3 (pHH3, mitotic cells) and BubR1, a protein that accumulates at unattached kinetochores and diffuses to the cytoplasm, to inhibit

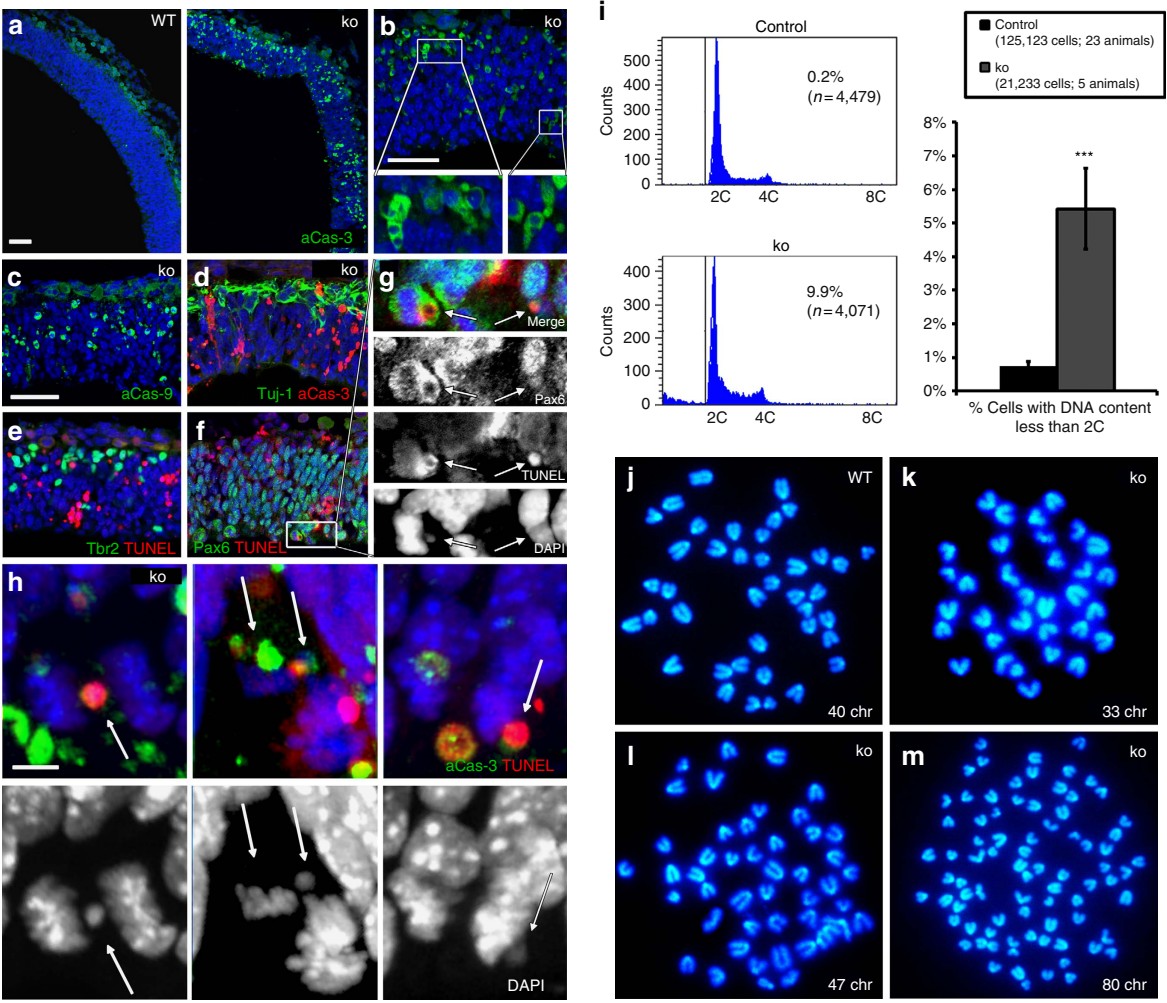

**Figure 4 | Cell death and aneuploidy in *Diaph3 ko* mice.** (**a–h**) E10.5 Cortical sections stained for aCas3 (**a,b**), aCas9 (**c**), aCas3 and Tuj1 (**d**), Tbr2 and TUNEL (**e**), Pax6 and TUNEL (**f,g**), and aCas3 and TUNEL (**h**). Widespread apoptotic profiles were found in the mutant (**a,b**). Immunoreactivity for aCas9 points to the mitochondrial apoptotic pathway (**c**). Co-staining with Tuj1 (neurons, **d**), Tbr2 (intermediate progenitors, **e**) and Pax6 (radial glial progenitor cells, **f**) identified the latter as the dying population. Apoptotic cells (TUNEL and/or aCas3, **g,h**) in the VZ showed asymmetric nuclear segregation (**g**), with in some cases the presence of aberrant nuclear profiles and micronuclei (arrows in **g,h**). Scale bars, 50 μm (**a–f**) and 5 μm (**h**). (**i**) Abnormal genomic content in mutant cortical progenitors. The DNA content is plotted in x axis and the number of cells (Count) in the y axis. The two main peaks represent DNA content of cells before (2C) and after (4C) S phase. A significant fraction of mutant cells had a DNA content less than 2C (5.43 ± 1.2% of mutant cells versus 0.73 ± 0.12% of control cells; n = 125,123 control cells from 23 animals and 21,233 ko cells from 5 animals, P < 0.0001; z-test). Error bars represent s.e.m. (**j–m**) Metaphase spreads prepared from WT (**j**) and mutant (**k–m**) neurospheres. Metaphases with less (**k**) and more (**l**) than forty chromosomes (chr) were frequent in mutant clones. Tetraploid metaphases were also occasionally observed (**m**).

progression to anaphase[24] (Fig. 5a–d). Unexpectedly, the percentage of BubR1[+]/pHH3[+] cells in pre-anaphase (that is, dividing cells that activate the SAC and arrest in metaphase) was decreased in the mutant tissue (WT: 65.05 ± 1.62%, *ko*: 53.26 ± 0.78%, n = 1,473 cells from 3 WT animals and 1,144 cells from 3 *ko* animals; P < 0.0001; Fig. 5e). Furthermore, the overall density of mitotic cells declined considerably (WT: 8.38 ± 0.29 cells per 100 μm of VZ, *ko*: 6.22 ± 0.23 cells per 100 μm of VZ, P < 0.0001; Fig. 5f), whereas the fraction of postmetaphasic cells among mitotic cells increased in *Diaph3 ko* tissue (WT: 27,8 ± 2.06%, *ko*: 39.91 ± 3.2%, n = 786 cells from 6 WT animals and 889 cells from 5 *ko* animals; Fig. 5g). These results indicate that mutant cortical progenitor cells are less sensitive to nuclear division errors and do not systematically halt in metaphase, in spite of chromosome mis-segregation. They also suggest that the stringency of the SAC is downregulated in *Diaph3 ko*. Given that haploinsufficiency of BubR1 is sufficient to cause aneuploidy in mice and humans[25,26], and that Diaph3

interactors APC and EB1 regulate the alignment and segregation of chromosomes through interaction with BubR1 (refs 27,28), we assessed the amount of the SAC protein BubR1. Western blot analysis of telencephalon extracts showed an ~50% reduction of BubR1 levels in mutant brain tissue (Fig. 5h,i). Moreover, BubR1 could be immunoprecipitated from cortical lysates of Diaph3 transgenic (TG) mice using anti-Diaph3 antibodies (Fig. 5j). To further explore the link between Diaph3 and BubR1, we infected Diaph3-deficient cortical progenitors with lentiviruses expressing either green fluorescent protein (GFP) or BubR1-IRES-GFP, cultured them as neurospheres for four passages and analysed their karyotype. Whereas the number of aneuploid metaphases was 60.5% in GFP-infected neurospheres, this percentage dropped to 33.9% in the BubR1-IRES-GFP-infected neurospheres (Supplementary Fig. 7), indicating that BubR1 overexpression reduced the number of aneuploid cells and rescued partially the Diaph3 phenotype. Conversely, we downregulated BubR1 by small interfering RNAs (siRNAs) in

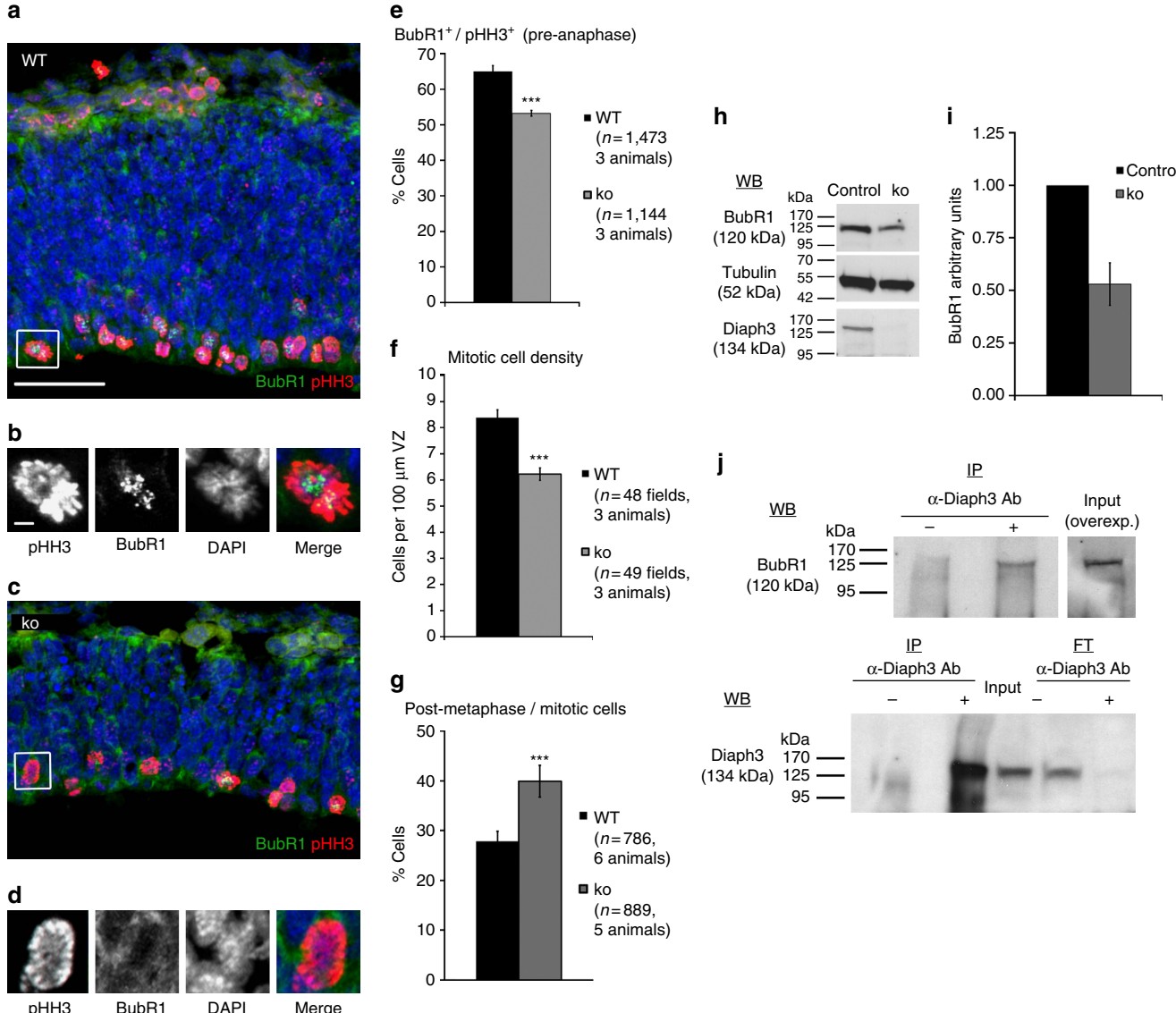

**Figure 5 | Lack of Diaph3 disables the SAC. (a–d)** Immunostaining for BubR1 (green) and pHH3 (red) of cortical sections from WT (**a**) and *Diaph3 ko* (**c**) E10.5 embryos. (**b,d**) Enlargements of the boxed areas in **a,c**, respectively. BubR1 (a hallmark of SAC activation) accumulated more in normal than in Diaph3-deficient cells (quantification in **e**; n = 1,473 control cells from 3 animals and 1,144 ko cells from 3 animals; P < 0.0001, z-test). (**f**) The density of mitotic cells in the cortex of E10.5 embryos was lower in the *ko* than in WT (n = 48 WT and 49 ko of 100 μm wide cortical stripes from 3 animals each genotype; P < 0.0001; Student's t-test). (**g**) Quantification of postmetaphasic cells in the population of mitotic cells. The percentage of mitotic cells that underwent the metaphase–anaphase transition was higher in the *ko* than in control littermates (n = 786 control cells from 6 animals and 889 ko cells from 5 animals; P < 0.0001, z-test). (**h,i**) Reduction of BubR1 protein levels in the mutant telencephalon detected by western blotting and quantified relatively to tubulin. The same membranes were blotted for Diaph3 to confirm the absence of the protein (**h**). The level of BubR1 decreased by half in the ko (**i**; n = 15 embryos in 4 pools for each genotype). (**j**) Western blotting (WB) detects BubR1 on IP with anti-Diaph3 antibodies from TG cortical lysates. Diaph3 was used as a positive control for the IP (Input). No signal was found in the eluted fraction (flow-through, FT) in presence of Diaph3 antibodies. Detection of the protein in the lysate (Input) required overexposure of the film. Scale bars, 50 μm (**a,c**) and 5 μm (**b,d**). Error bars represent s.e.m.

WT neurospheres and did not detect any change in Diaph3 levels by western blotting (Supplementary Fig. 8a). Knockdown of BubR1 in mouse embryonic fibroblasts (MEFs) derived from TG embryos overexpressing Diaph3 and control littermates led to a substantial loss of MEFs in both genotypes. In addition, the percentage of aneuploid metaphases was 42% in control MEFs and 44.6% in Diaph3 TG MEFs, suggesting that the overexpression of Diaph3 was not sufficient to prevent aneuploidy and cell death induced by BubR1 downregulation (Supplementary Fig. 8b,c). We used siRNA to knock down the human DIAPH3 in in the hypotriploid HEK293T cells[29] (Supplementary Fig. 9). Karyotype analysis showed that in cells treated with control siRNAs, 65% of metaphases had a chromosome number ranging from 50 to 84, 15% had between 21 and 49 chromosomes, and 19% had < 20 chromosomes. In DIAPH3-siRNA-treated cells, 27% of metaphase spreads had between 50 and 70 chromosomes, 19% had between 21 and 49 chromosomes, and 49% had < 20 chromosomes. Remarkably, DIAPH3 downregulation resulted in a concomitant reduction of BUBR1 levels, suggesting that, similar to that in mice, the downregulation of DIAPH3 in human cells disturbs karyokinesis and promotes aneuploidy. Collectively, these results establish a relationship between Diaph3, BubR1 and chromosome segregation during division of cortical progenitors.

Key to the SAC activity is the CPC, formed by Aurora kinase B, Survivin, Borealin and Incenp. CPC is located on centromeric DNA and 'senses' the tension between sister chromatids at the kinetochore[30]. In case of improper connections, Aurora B phosphorylates kinetochore proteins, thereby destabilizing spindle MT–kinetochore interactions[31,32]. To explore the relationship between Diaph3 and CPC in cortical progenitors *in vivo*, we examined the subcellular localization of the Diaph3 protein on *in utero* electroporation of a plasmid coding for a GFP::Diaph3 fusion protein[3] in E13.5 embryos. One day after electroporation, the GFP was present in dorsal telencephalic cells. The signal was perinuclear and extended to apical and basal processes in non-dividing cells (Supplementary Fig. 10, arrowheads)[33] and more diffusely distributed around the chromatin in dividing cells (Supplementary Fig. 10, arrows)[34,35]. We carried out co-immunoprecipitation (IP) experiments on E13.5 brain extracts from Diaph3 TG mouse[11] and detected a previously uncharacterized interaction between Diaph3 and Survivin (Fig. 6a). Thus, Diaph3 interacts with CPC components during division of cortical progenitor cells. As Diaph3 was shown to form a complex with APC and EB1 at the plus-end of MT[17], and APC and EB1 control the centromeric localization[36] and stability[37] of the CPC component Aurora B, we studied the

localization of CPC proteins in mutant and control neural progenitors. In dividing WT progenitors, CPC proteins relocate to the midzone after chromosome separation, to prevent SAC activation due to loss of tension[38]. In agreement, we found that CPC protein Survivin localized at the midzone in most dividing cells beyond the stage of metaphase ($86.61 \pm 1.40\%$, $n = 193$ cells). By contrast, a substantial fraction ($44.68 \pm 2.41\%$, $n = 288$ cells) of post-metaphasic mutant cells displayed an atypical distribution of Survivin (Fig. 6b,c).

## Discussion

In this work, we generated and analysed a mouse line carrying a mutation in the *Diaph3* gene. In the *ko* mice, cortical progenitor cells undergo apoptosis as early as E10.5. Using flow cytometry analysis, we found a sevenfold increase in the proportion of aneuploid cells in the mutant telencephalon (5.43% versus 0.73%). These cells presumably die, depleting progressively the population of progenitors and leading to cortical hypoplasia, as shown by the marked reduction in all cortical cell types in *Diaph3 ko* embryos at E13.5. Aneuploidy could eventually give rise to neoplastic transformation. Remarkably, mutations in the human *DIAPH3* gene are frequently found in metastatic cancers and downregulation of *DIAPH3* increases metastatic invasion in xenotransplanted mice[39,40]. The nuclear asymmetric division we report here could increase chromosomal instability, promoting the emergence of new mutations and facilitating the acquisition of metastatic properties.

Karyotype analysis of neurospheres derived from *Diaph3 ko* NE cells revealed that 65% of mutant cells had chromosome numerical abnormalities. The high prevalence of aneuploidy in neurosphere cultures was unexpected in view of the flow cytometry results (65% *in vitro* versus 5% *in vivo*). However, several factors can account for this apparent discrepancy. First, cells with 38, 39, 41 or 42 chromosomes, which are the most abundant among the aneuploid cell population, are not easily distinguishable by flow cytometry, as their DNA content is close to that of euploid cells. Second, aneuploid cells with a DNA content higher than 2C cannot be discriminated from euploid cells in S phase, which also have a DNA content between 2C and 4C, and were not taken into account in the flow cytometry analysis. Third, contrary to the developing brain, where aneuploid cells are mostly eliminated by apoptosis, such cells accumulate *in vitro*, probably because culture conditions promote their viability. Interestingly, the incidence of aneuploidy in *Diaph3*-deficient neurospheres is comparable to that of cultured MEFs carrying a hypomorphic mutation of BubR1 (ref. 41). Even though tetraploidy was associated with other chromosome numerical abnormalities as in the mosaic variegated aneuploidy[26], we cannot exclude the possibility that tetraploid cells arose from cytokinesis failure. Diaph3 has been associated with actin dynamics *in vitro*[4,9,10]. *In vivo*, the lack of Diaph3 disrupts the accumulation of filamentous actin in the contractile ring, impairing cytokinesis of erythroid cells and generating multinucleated cells. We did not observe overt modifications of actin cytoskeleton in neuroepithelial cells in *Diaph3 ko* at early embryonic stages, suggesting that depletion of Diaph3 may not affect all cells similarly. The fact that early erythroid precursors are not affected by the mutation[10] supports this assumption. In those cells, similar to that in cortical progenitors, Diaph proteins are coexpressed and could have redundant functions in the formation of the contractile ring.

Diaph3 co-immunoprecipitates with the mitotic spindle protein BubR1 and its mutation reduces by half the overall level of BubR1. We did not see any accumulation of cells in metaphase. Thus, the lack of Diaph3 weakens the spindle checkpoint and behaves as a BubR1 hypomorphic (h) allele. In support of this, the phenotype of *Diaph3 ko* phenocopies that of *BubR1^h* mice in

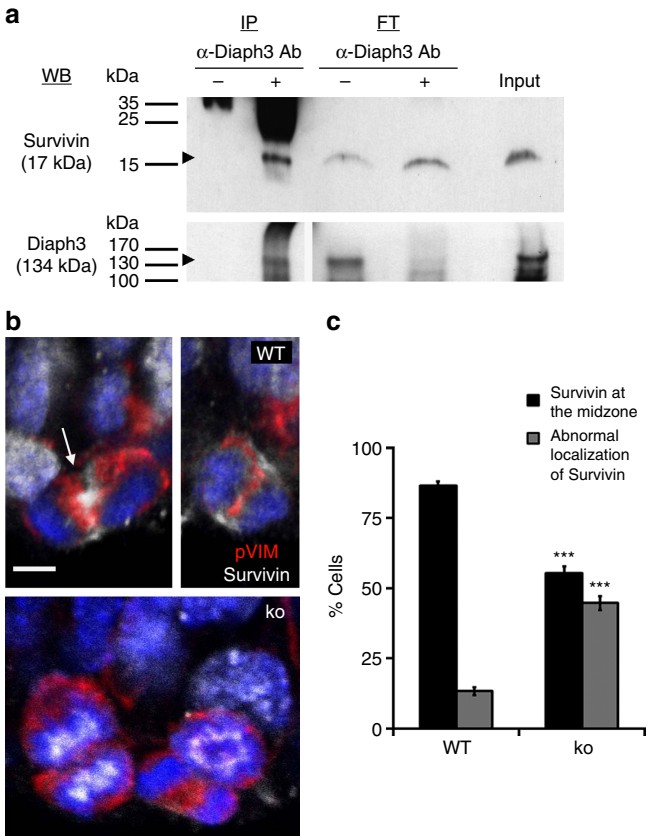

**Figure 6 | Diaph3 interacts with Survivin. (a)** Co-immunoprecipitation of endogenous Survivin and Diaph3 from brain lysates of *Diaph3* TG mice. Western blotting was positive for Survivin on anti-Diaph3 immuno-precipitation ( + ). No signal was present in absence of antibodies ( − ). Western blotting for Diaph3 was used as a positive control. FT, flow-through. **(b)** Unlike in WT postmetaphasic cells, where Survivin is located at the midzone (arrow), this localization was disturbed in the Diaph3 ko. **(c)** Quantification of postmetaphasic cells with normal or abnormal distribution of Survivin ($n = 193$ control cells and 288 ko cells; $P < 0.0001$, z-test). Error bars in **c** represent s.e.m. Scale bar, 10 μm.

which mitotic slippage, formation of micronuclei, premature chromatid separation, aneuploidy and decreased number of mitotic cells were described[25,41]. A link between BUBR1 and chromatid separation was also reported in patients with mosaic variegated aneuploidy, a rare disease associated with intrauterine growth retardation, aneuploidy, microcephaly and hydrocephalus[26,42], further supporting the Diaph3-BubR1-nuclear division axis.

What could be the connection between Diaph3 and BubR1? *In vitro* studies have shown that Diaph3 stabilizes MTs in a APC–EB1-dependent manner[15,17,43]. On the other hand, the APC–EB1 complex controls alignment and segregation of chromosomes[28,44] through interaction with BubR1 (ref. 27). The APC–EB1 complex is also required for centromeric localization of the CPC effector Aurora B[37]. CPC 'senses' the tension between sister chromatids at the kinetochore[30]. In case of improper connections, Aurora B phosphorylates kinetochore proteins, which destabilizes spindle MT–kinetochore interactions[31,32]. Hence, Diaph3, APC, EB1, BubR1 and CPC proteins belong to a molecular toolkit that regulates interactions between the kinetochore and spindle MTs, as well as the tightness of the mitotic checkpoint.

In addition to the cortical phenotype, *Diaph3* mutant embryos display growth retardation, twisting neural tube, facial deformities and increased number and size of brain blood vessels. More than 97% of mutants die before E14.5, most probably from a severe anaemia due to loss of erythroid cells. Animals that survive until young adulthood exhibit smaller brain, hydrocephalus and growth retardation. These features are common findings in mouse models of microcephaly[45,46] and in patients with type II microcephalic osteodysplastic primordial dwarfism[47], whose life is imperiled by modifications of cerebral blood vessels often resulting in stroke or aneurysm. The severity of the Diaph3 phenotype (that is, embryonic lethality) might have precluded the establishment of a causal link between microcephaly and DIAPH3 loss-of-function mutations. Although mutations in DIAPH1 were associated with microcephaly in humans[48,49], neither *Diaph1* or *Diaph2* single *ko*, nor *Diaph1* and *Diaph2 dko* have symptoms of primary microcephaly[8,48]. The involvement of Diaph3 in microcephaly suggests a functional redundancy and/or divergent roles of *Diaphanous* genes in humans and mice.

In conclusion, our results provide evidence that Diaph3 protects cortical progenitors against mitotic error-induced apoptosis, by preserving the activity of the spindle checkpoint (see model in Fig. 7). Loss of Diaph3 function does not trigger nuclear division errors in the strict sense. Such events occur physiologically, especially in fast dividing cells such as mammalian cortical progenitors. Rather, the lack of Diaph3 loosens the spindle checkpoint enabling a fraction of aberrantly dividing cells, which normally halt in metaphase until nuclear segregation is properly completed, to 'slip' into anaphase, causing aneuploidy and /or mitotic catastrophe.

## Methods

**Animals.** All procedures were carried out in accordance with European guidelines and approved by the animal ethics committee of the Université catholique de Louvain. Mouse lines used in this study were: *Diaph3 ko* and *Diaph3^f/f* (Supplementary Fig. 1a), *Emx1-Cre* (B6.Cg-Emx1^tm1(cre)Krj/J; Jackson Lab)[22] and FVB-Tg(CAG-Diaph3)924/Lesp/J (Jackson Lab). For Hypoxyprobe staining, *Diaph3* heterozygous timed pregnant females were injected with 60 mg kg$^{-1}$ pymonidazole hydrochloride and E10.5 embryos were harvested and fixed after 45 min. As a positive control, we injected subcutaneously $1 \times 10^6$ B16F10 cells in a C57BL/6J mouse and collected the tumour 10 days after injection.

**In situ hybridization.** A *PCRII–Topo Diaph3* plasmid (Addgene #45602) was used to produce a UTP[33]-labelled probe as described previously[50,51]. Coronal and sagittal cryosections from E10.5 and E13.5 embryos, and P0 brains were treated with 1 μg ml$^{-1}$ proteinase K in 0.1 M Tris HCl pH 8 and 10 mM EDTA, rinsed in diethyl pyrocarbonate (DEPC)-treated water and acetylated for 10 min at room temperature in 0.25 M acetic anhydride–0.1 M triethanolamine. Slides were incubated overnight at 65 °C in a humid chamber with denatured probes (1 μg ml$^{-1}$) in hybridization solution (50% formamide, 10% dextran sulphate, 0.3 M NaCl, 20 mM Tris HCl pH 7.5, 5 mM EDTA, 1 × Denhardt's solution, 0.6 mg ml$^{-1}$ yeast transfer RNA and 0.1% SDS). Slides were washed for 30 min at 65 °C in 50% formamide–2 × SSC, rinsed in 2 × SSC and treated for 1 h at 37 °C with 1 μg ml$^{-1}$ RNASe in NTE buffer (0.5 M NaCl, 10 mM Tris HCl pH 7.5 and 5 mM EDTA). Slides were washed in 2 × SSC and 0.2 × SSC at 65 °C for 1 h each, dehydrated with ethanol and exposed to Kodak Biomax Maximum Resolution films (Sigma).

**Immunohistochemistry.** For histology, E12.5 and E13.5 were fixed in Bouin's solution and processed for haematoxylin–eosin staining. For immunohistochemistry, embryos were fixed in 4% paraformaldehyde. Cryosections (16–18 μm thick) were processed for antigen retrieval in 0.01 M sodium citrate (pH 6) for 5 min, washed with PBS and blocked in PBS supplemented with 0.3% Triton X-100 and 5% normal goat serum. Slides were incubated overnight at 4 °C in the same blocking solution containing primary antibodies, washed and incubated with appropriate Alexa Fluor-conjugated secondary antibodies (Invitrogen). Primary antibodies were as follows: Pax6 (Covance; catalogue number PRB-278 P; 1:100), Tbr2 (Abcam; catalogue number ab23345; 1:500), Tbr1 (Abcam; Cat catalogue number ab31940; 1:500), active Caspase-3 (Cell Signaling; catalogue number 9661; 1:500), active Caspase-9 (Cell Signaling; catalogue number 9509; 1:500), Survivin (Cell Signaling; catalogue number 2808; 1:400), phospho-Vimentin (Abcam; catalogue number ab22651; 1:50), Doublecortin (Cell Signaling; catalogue number 4604; 1:400), pHH3 (Cell Signaling; catalogue number 9701; 1:100), Tuj-1 (Covance, catalogue number MMS-435 P-0250; 1:1,000), GFP (AVES; catalogue number GFP-1020; 1:1,000), Aurora-B (Abcam; catalogue number ab2254; 1:500), Hp-1 (Hypoxyprobe, catalogue number HP-100 mg), BubR1 (BD Bioscience; catalogue number 612502; Clone 9, 1:500) and isolectin GS-IB4-Alexa Fluor 568 Conjugate (Invitrogen; catalogue number I21412; 1:500). TUNEL staining was performed using an *In Situ* Cell Death Detection Kit (Roche, catalogue number 11684795910) according to the manufacturer's instructions. Nuclei were counterstained with DAPI. Images were acquired with an Olympus FV1000 confocal microscope and edited with Photoshop (Adobe Systems). Quantification of stained sections was performed on pictures generated with the same x, y and z size. Z-size is set to 3 μm in Fig. 1 and 5 μm in Fig. 4.

**DNA content analysis.** Cortical progenitor cells were collected from E11.5 brains and dissociated to single cells using a Neural Tissue Dissociation kit (MACS-Miltenyi Biotec) according to the manufacturer's guidelines. Cells were resuspended in PBS, fixed in ice-cold 70% ethanol for 30 min on ice and treated with propidium iodide for 30 min at 37 °C. Analysis was performed with a FACSCanto Flow Cytometer (BD Biosciences).

**Culture of embryonic neural progenitor cells.** Cortices were isolated from at E10.5 embryos in glucose enriched L-15 and mechanically dissociated in cold PBS. The cells were spun down and suspended in DMEM-F12 supplemented with GlutaMAX, non-essential amino acids, penicillin–streptomycin, pyruvate, B27 minus Vitamin A, N2 (Life Technologies), 10 ng ml$^{-1}$ basic fibroblast growth factor and 20 ng ml$^{-1}$ epidermal growth factor (R&D System) in low cell-binding 96-well round-bottom plates (Sigma). Neurospheres were dissociated to single cells every 4 days using TrypLE (Gibco). Cells were counted and plated in 12-well Cellstar cell culture plates (Greiner) at a density of 150,000 cells per well. To ascertain the neural identity of neurospheres, cells were plated on 12 mm poly-D-lysine-coated round coverslips (Corning) in 24-well CellStar cell culture plates (Greiner) at a density of 50,000 cells per well. After 4 days, cells were fixed for 15 min in 2% paraformaldehyde in PBS, rinsed in PBS, blocked in 5% normal goat serum, 0.3% Triton X-100/PBS for 20 min and incubated overnight at 4 °C in blocking buffer containing mouse anti-Nestin (1:1,000; Millipore). Cells counterstained with DAPI. Coverslips were mounted with Mowiol.

**Preparation of metaphase spreads.** Neurospheres (passage 8) were cultured with 0.05 μg colcemid per ml for 3 h, pelleted by centrifugation for 5 min at 1,000 r.p.m., treated with hypotonic solution (0.075 M KCl in water) and fixed in 3:1 methanol:acetic acid. Metaphase chromosome spreads were obtained by releasing a few drops of cell suspension onto sulfuric acid (25%)-cleaned slides. The slides were mounted with Mowiol medium with 0.25 μg ml$^{-1}$ DAPI and examined with an Axioskop fluorescence microscope (Zeiss, Germany).

**In utero electroporation.** The *pEFmEGFP-mDia2* plasmid (Addgene #25407), coding for a GFP::Diaph3 fusion protein[3] was introduced in neural progenitor cells via *in utero* electroporation. E13.5 embryos were injected intraventricularly with 1 μg DNA, electroporated according to standard protocols and collected 24 h later.

**Western blotting.** Tissues were homogenized in lysis buffer containing 50 mM Tris HCl pH 7.5, 150 mM NaCl, 1 mM EDTA, 1% Triton-X 100 and protease inhibitors (Roche). Cell lysates were centrifuged at 13,000 g for 15 min at 4 °C.

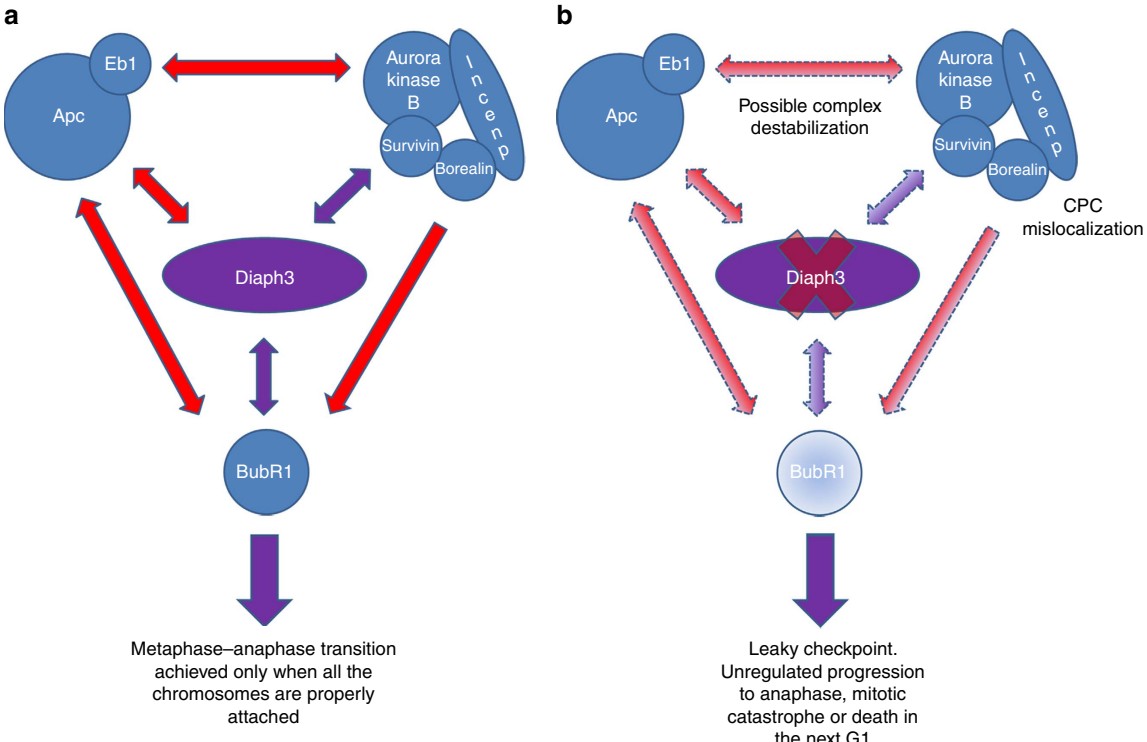

**Figure 7 | Working model of Diaph3 function in the spindle checkpoint. (a)** Molecular interactions between Diaph3, CPC and SAC proteins during cell division. Documented interactions are depicted in red and new findings are shown in purple. Diaph3 localizes with CPC proteins and co-immunoprecipitates with Survivin and BubR1. (**b**) Absence of Diaph3 probably disrupts the Diaph3–APC–Eb1 complex, which interacts with and stabilizes the CPC complex at the spindle–kinetochore interface. Loss of Diaph3 impairs the localization of CPC proteins and the ability of dividing cells to activate the SAC. Diaph3-deficient cells fail to accumulate BubR1, leading to slippage to anaphase and mitotic catastrophe and/or cell death of aneuploid progeny.

Protein quantification was performed with a BCA kit (Pierce). Supernatant was mixed with 5 × SDS-loading buffer and heated at 95 °C for 5 min. Equal amount of proteins were loaded on 10 or 4–20% TGX gels (Bio-Rad), separated by SDS–PAGE and transferred to nitrocellulose membranes (GE Healthcare). Membranes were blocked with 5% fat-free dry milk and incubated overnight at 4 °C with rabbit anti-Diaph3 C-terminal (1:5,000)[3], mouse anti-BubR1 (BD Bioscience; catalogue number 612502; Clone 9, 1:500), mouse anti-alpha-tubulin (Sigma; catalogue number T 6199; Clone DM1A; 1:2,000) or chicken anti-GAPDH (Millipore; AB2302; 1:2,000). Relative BubR1 protein quantification was performed using the ImageJ Gel analyser tool. Values were normalized to GAPDH or α-tubulin. The amount of BubR1 protein in controls was set to one. Images of western blottings have been cropped for presentation. Full-size images are presented in Supplementary Fig. 11.

**Immunoprecipitation.** Brain or cortical lysates obtained from *FVB-Tg(CAG-Diaph3)924/Lesp/J* mice, which constitutively overexpress Diaph3, were prepared as described above. Proteins were quantified using the BCA kit (Pierce). Two hundred micrograms of proteins were incubated overnight in lysis buffer containing protease inhibitors and 5 μg rabbit anti-Diaph3 antibody. As negative control for the IP, the lysate was treated in the same conditions but without antibody. Immune complexes from anti-Diaph3 and control IP were recovered by incubation (2 h at 4 °C) with 20 μl protein A-coated agarose beads (Invitrogen) and centrifugation (2 min at 2,500 g). Beads were washed three times in PBS and proteins were eluted by addition of 5 × Laemmli buffer and heating at 95 °C for 5 min. The samples were processed for western blotting, for Survivin and BubR1 proteins. Total brain or cortical lysate were used to immunoprecipitate Survivin and BubR1, respectively.

**Lentiviral vector preparation and infection.** Lentiviral particles have been prepared as reported[52]. Briefly, HEK293T cells (ATCC CRL-3216) were transiently transfected with Lipofectamine 2000 (Invitrogen) using a third-generation system with lentiviral pHIV transfer vectors together with packaging plasmids pENV, pMDL and pREV. pHIV-IRES-EGFP and pHIV-BubR1-IRES-EGFP transfer plasmids were used to generate lentiviral particles. The first was produced by replacing the dTomato cDNA of a pHIV-dTomato vector (Addgene plasmid #21374) with an IRES-EGFP cassette from a pIRES2-EGFP plasmid (Clontech). The second was obtained following the same strategy, but after insertion of the mouse BubR1 ORF (Origene) upstream of the IRES-EGFP. Twenty-four and 48 h

after transfection, the virus containing supernatant was harvested, concentrated using 50 kDa Amicon filters (Millipore), titrated and applied to dissociated Diaph3 ko neurospheres for 48 h. Cells were then rinsed several times with PBS and fresh medium was added. Fluorescent neurospheres infected with IRES-EGFP (Control) or BubR1-IRES-EGFP (Rescued) were amplified for four passages, imaged with a Zeiss inverted microscope Axiovert.A1 and processed for karyotype analysis.

**BubR1 downregulation in MEFs.** MEFs from *Diaph3* TG and WT littermates were isolated following standard protocols. Briefly, E13.5 embryos were eviscerated, minced in small pieces, trypsinized for 30 min and mechanically dissociated. Cells were cultured in ko-DMEM, supplemented with GlutaMAX, non-essential amino acids, penicillin–streptomycin, pyruvate and 10% fetal bovine serum (FBS; Invitrogen). MEFs from 3 *Diaph3* TG and 3 controls were pooled together, seeded in 12-well plates at 50–70% confluency and cultured for 24 h. Cells were then treated for 48 h with 1 μM of either Accell Control siRNA (5′-UGGUUUACAUG UCGACUAA-3′) or Accell mouse BubR1 Smart Pool siRNAs (5′-CCACUAAG CUCGAAUCCUA-3′; 5′-CUACUAGAAUUAAGUGCUU-3′; 5′-CCUAUGACU AUGUAAAUAA-3′; 5′-GCUUUUACUCUGGAGAUGA-3′; Dharmacon) diluted in Accell Delivery Medium. The medium was replaced with complete medium and cells were grown for 24 h, dissociated with TrypLE and seeded again in 12-well plates. After three cycles of knockdown, MEFs were processed for western blotting or karyotype analysis.

**DIAPH3 knockdown in 293T cells.** HEK293T cells (ATCC CRL-3216) were cultured in DMEM supplemented with GlutaMAX, penicillin–streptomycin and 10% FBS (Invitrogen), amplified in 25 ml flasks (Greiner) and seeded in 24-well plates (Greiner) at a confluency of 50–70%. Cells were treated for 48 h with 1 μM of either Accell Control siRNA (5′-UGGUUUACAUGUCGACUAA-3′) or Accell human DIAPH3 Smart Pool siRNAs (5′-CUACAAGCUUUUAAGUCUC-3′; 5′-GGAUUUGCUUUGUAAACUU-3′; 5′-GCGUUUAUUAGAAAUGAAG-3′; 5′-CUGAUAUACUGAAUUUUGU-3′; Dharmacon) diluted in Pro293a-CD medium (Lonza) supplemented with GlutaMAX and 2% FBS. Medium was replaced with complete medium and cells were grown for 24 h, dissociated with TrypLE and seeded again in 12-well plates. Effective DIAPH3 knockdown was checked twice at passage 2 and 3 via western blot analysis. After four cycles of knockdown (16 days of continuous DIAPH3 downregulation), HEK cells were processed for karyotype analysis.

**Statistics.** For analysis of different conditions or genotypes, the Student's *t*-test was used. For CPC mislocalization, BubR1 relative amount, postmetaphasic fraction of mitotic NE cells and flow cytometry data, the Fisher's exact test and *z*-test were used. Plotted data are represented as mean ± s.e.m. Significance levels are as follows: *$P < 0.05$, **$P < 0.01$ and ***$P < 0.001$.

**Data availability.** Data sharing not applicable to this article. All data were included in the main manuscript and Supplementary Information, and no data sets were generated during the current study.

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

## Acknowledgements

We thank Paolo Porporato for xenografts and help with production of lentiviral particles; Yves Jossin and members of the Developmental Neurobiology group for discussions; and Valérie Bonte, Rachid El Kaddouri, Isabelle Lambermont and Esther Paitre for technical support. This work was supported by the following grants: FNRS PDR T0002.13, FNRS

PDR T00075.15, Interuniversity Poles of Attraction (SSTC, PAI p6/20 and PAI7/20), Fondation Médicale Reine Elisabeth, Fondation JED-Belgique and WELBIO-CR-2012A-07, all from Belgium. F.T. is a senior research associate of the Belgian Funds for Scientific Research (FRS-FNRS).

## Author contributions

D.D. performed most of experiments. F.T. produced *Diaph3 ko* mice and carried out *in situ* hybridization and chromosome content analysis. D.D., A.M.G. and F.T. designed research. A.A. provided fundamental reagents. D.D. and F.T. wrote the manuscript. All authors edited and approved the manuscript.

## Additional information

**Competing financial interests**: The authors declare no competing financial interests.

**How to cite this article**: Damiani, D. *et al.* Lack of Diaph3 relaxes the spindle checkpoint causing the loss of neural progenitors. *Nat. Commun.* **7**, 13509 doi: 10.1038/ncomms13509 (2016).

**Publisher's note**: 

