## [Peer Review File · Nature Communications]

Reviewers' comments:

Reviewer #1 (Remarks to the Author):

Diap3 has a role in the actin cytoskeleton and this manuscript provides novel information on its importance for cortical development by examining the effects of mutating it. The effects are profound. Cell divisions and death are affected adversely by its loss and microcephaly results.

Expression of Diap3 is shown - only low magnification, so no idea can be gained about whether all progenitors express it or if it is VZ or SVZ or both, it says it's in RGCs but this is not obvious with this scale - is it in IPCs? Please show high magnification with appropriate stains and markers to show what cell types and layers it's expressed by. It appears to be expressed at least as strongly outside the nervous system, please comment on this and the implications for non autonomous effects.

Diap3 is then inactivated and this has striking and powerful effects. Numbers of progenitors of different types and neurons were counted in different cortical regions, but these are in fact densities per unspecified field, which is meaningless since we don't know what a field is. It looks like the thickness of the cortex is reduced but density of cells is roughly normal. You need to give a better description of what the changes are in terms that are defined by standard scientific values such as microns, mm, etc.

This journal wants you to define the meaning of your error bars in the legend and you don't. You do say it in Methods. The tests used seem appropriate to me.

Page 8, line 4-5, I think there's a typo in the first genotype given, isn't this a control?

The attempts to show the effect of deletion is autonomous are ingenious and fairly convincing, although they don't rule everything out. They do show the effects are internal to the cortex and the overall hypothesis advanced to explain the findings is very reasonable.

Overall, I think this will be of great interest to a broad range of readers and certainly it is very innovative. If the points above can be addressed thoroughly it will be an even better paper, in my opinion. It will influence thinking in this area.

The manuscript is clear and generally well written.

Reviewer #2 (Remarks to the Author):

Comments on "Lack of Diap3 relaxes the spindle checkpoint causing the loss of neural progenitors and microcephaly", by Damiani et al.

Damiani and colleagues propose that Diap3, one of 3 Diaphanous isoforms, is a key member of the regulatory pathway that decides whether neural progenitors in the developing mouse cortex should exit mitosis. For this, they generated a conditional knock-out mouse, in which they show severe and broad developmental defects and death before birth. In the cortex, they show significant cell death in neural progenitors, which they associate convincingly to severe defects in chromosome segregation. Establishing a strong causal link would, however, need some more data. The potentially most interesting point of the manuscript is the link to the SAC protein BubR1, which seems impaired from enforcing a normal spindle checkpoint in the absence of Diap3. The data for the association with CPC proteins, on the other hand, is weak. A revised manuscript could be very interesting, but also more suited to a specialized journal.

Main points:

1- This manuscript focuses on defects in the developing cortex, yet others have shown, and the authors confirm, that Diap proteins are also broadly required elsewhere, with some functional and mechanistic overlap with what the authors observe in neural progenitors. In this light, a more

specialized journal seems a better fit for these more specific data.

2- The full Western blots, including e.g. markers and loading controls, must be included in all figures, as key conclusions suggested by the authors rely heavily on them.

3- Even though a direct causal link between aneuploidy and cell death for Diap3 ko is implied, likely, and would be a key part of the study, the data for this should be more robust. One possibility would be to quantify what is shown in Fig. 4h, mitotic cells with/without segregation errors and positive/negative for cell death markers.

4- The validity of the data in Fig. 6 depends fully on the GFP-Diap3 fusion protein, yet validation data for this construct is completely missing. Likewise, it is unclear if overexpression per se, and overexpression differences between cells could be affecting these results (also those in Sup. Fig. 6). Therefore, little can be robustly claimed from these data about the localization and associated functionality of Diap3.

5- Even if the construct and expression levels were found to be functional, I see no convincing "colocalization" of GFP-Diap3 with CPC components in Fig. 6 a-b. If anything, these images argue for a simple exclusion of GFP-Diap3 from the chromatin volume. A proper colocalization analysis, quantifying signals in higher resolution stacks and showing single channels would be necessary. See also point 2.

Minor points:

6- The quality of the images provided should be higher, although I realize this could be due to the editorial processing of the original pictures.

7- It is puzzling that the difference in Fig. 5e is deemed statistically significant, but that in Fig. 5g is not. A revision of the samples, independent sample sizes and/or statistics would seem necessary.

Reviewer #3 (Remarks to the Author):

Comments to the author

In this paper, the authors investigate the role of the protein Diap3 in mouse development. Diap3 is known to regulate cytokinesis, but instead they found that Diap3 is a regulator of chromosome segregation in mouse cortical progenitor cells. Loss of function of Diap3 led to massive apoptosis of neural progenitors during embryogenesis and microcephaly. The authors provide evidence for the interaction of Diap3 and BubR1, a regulator of the spindle assembly. They suggest a new molecular model highlighting the protective role of Diap3 against mitotic error induced apoptosis. The results are original and describe new functions of the Diaphanous formins, supporting Diap3 implication in primary microcephaly.

The authors have a longstanding experience in neural development, neuronal migration, cell polarity and cell division. The data and the manuscript are well presented, and the results provide proof consistent with the author's hypothesis: lack of Diap3 leads to loosening of the spindle checkpoint, enabling aneuploidy and consequent apoptosis in cortical progenitors; however, the manuscript could be improved after addressing the following major and minor changes:

MAJOR Changes:

1) The role of Diap3 in mouse cortical neurons is supported by the phenotypic data in the mutants, and suggests an important role in primary microcephaly. Relevant to the mutant phenotype is the interaction between Diap3 and BubR1, In this study, the authors show that in Diap3 mice, levels of BubR1 are reduced by 50%. In fact, BubR1 ko mice display some phenotypic features found in the Diap3 mice, such as progressive aneuploidy and dwarfism (Baker et al., 2004). However, the authors do not provide a rescue experiment that would point out if the aneuploidy and the small body size (including microcephaly) are a primary effect due to Diap3 absence or if they are due to the consequent reduction in BubR1 levels. The rescue experiment could be performed with in utero electroporation (E13.5) of BubR1 cDNA, or by transfection of BubR1 into Diap3 ko-derived neurospheres, for example. The interaction among Diap3 and BubR1 was not known at the time of the BubR1 ko mouse characterization. Thus it would be helpful to test whether Diap3 levels are lower in BubR1 ko mouse cortical neuroprogenitors as well. If Diap3 levels are lower in BubR1 ko mice compared to WT, rescue experiments transfecting Diap3 into BubR1 ko mice or neurospheres should be performed. The results would clarify if the phenotype is depending solely on individual gene loss or if it is due to a combined effect of the loss of both genes.

2) Overexpression of DIAPH3 in humans has similar effects of Diap3 overexpression in mouse, as cited by the authors (Schoen et al., 2010). However, there are no known cases of DIAPH3 downregulation in humans, probably due to its high lethality (as shown here in the mouse model). It would be important to test if DIAPH3 depletion in human cell lines leads to progressive aneuploidy and consequent apoptosis. There are modified HEK293 cell lines available with stable knock down of DIAPH3 that would be suitable for this purpose (Panoply[®] Human DIAPH3 Knockdown Stable Cell Line), especially due to the fact that HEK293 cells are usually hypotriploid. A reduction of chromosome number in these cells upon knockdown of DIAPH3, as well as increased apoptosis would strongly support an unidentified role for the Diap3 human homolog in miscarriages and rare cases of primary microcephaly.

MINOR Changes:

1) Although it is clear that Diap3 ko mice are smaller than their littermates (Fig 1d-e), the authors do not provide their body weight and the relative head size. It is important to provide this data to discriminate between general dwarfism and primary microcephaly, and to discriminate among the overlapping phenotypic features in the BubR1 ko mouse.

2) Page 8, line 4: there is a typo in the genotype of the mouse. In the sentence "Whereas no apoptosis was seen in Diap3f/f;Emx1-Cre or Diap3f/f cortices, Diap3f/f;Emx1-Cre embryos exhibited many apoptotic cells, demonstrating that cell death is caused by the lack of Diap3 in cortical progenitors rather than in blood cells (Supplementary Fig. 4) should be written as "Diap3f/+;Emx1-Cre", based on Supplementary Fig.4.

3) Discussion paragraph, page 12, line 4: more emphasis should be given to the sentence "These cells presumably die, depleting progressively the population of progenitors and leading to cortical atrophy (we observed a marked reduction in all cortical cell types in Diap3 ko embryos at E13.5)", as it contains part of the relevant findings of the paper. Instead, the sentence should be written as: "These cells presumably die, depleting progressively the population of progenitors and leading to cortical atrophy, as shown by the marked reduction in all cortical cell types in Diap3 ko embryos at E13.5."

4) Figure 4, page 27: Images for immunostaining of Tbr2/TUNEL, Pax6/TUNEL, aCas3/TUNEL are only provided for Diap3 ko cortical sections, but not for wild type cortical sections. These could be added, together with additional representative images of the same immunostainings in Diap3 cortical

sections, in the Supplementary material.

5) Figure 6, page 31: Figures a and b show two examples of mitotic cells per immunostaining, however only one of the example per immunostaining is labeled. It would be preferable to label the second example as well, creating a "a1-a2/ b1-b2" figure, or labeling the markers in the second examples as well. It would be worth pointing out that these figures refer to wild type mouse electroporated in utero.

6) Supplementary information, table 1: to my understanding, the "examined" column only accounts for cells analyzed that showed chromosomal number aberration, and not the total number of examined cells (which should be 142 and 263, respectively). The addition of a column to list the total number of analyzed cells would help clarify this point. I would also suggest to add the number of chromosomes for aneuploid cells in the header (i.e. 20).

Reviewers' comments:

Reviewer #1 (Remarks to the Author):

Diap3 has a role in the actin cytoskeleton and this manuscript provides novel information on its importance for cortical development by examining the effects of mutating it. The effects are profound. Cell divisions and death are affected adversely by its loss and microcephaly results. Expression of Diap3 is shown - only low magnification, so no idea can be gained about whether all progenitors express it or if it is VZ or SVZ or both, it says it's in RGCs but this is not obvious with this scale - is it in IPCs? Please show high magnification with appropriate stains and markers to show what cell types and layers it's expressed by.

We thank the reviewer for her/his overall positive evaluation of our work. To refine the expression of Diaph3, we used the RNAscope technology that allows a cellular localization. We also used Pax6, Tbr2 and Doublecortin immunostaining, as suggested by the reviewer. We found that Diaph3 mRNA is present mostly in the ventricular zone, where Pax6 progenitors reside. Co-localization with Tbr2 is minimal if existing, and no co-expression with immature neurons was observed. Interestingly, Diaph3 seems to be expressed in a cell cycle-dependent manner, as the signal is higher in the outermost ventricular zone where the S phase occurs during interkinetic nuclear migration. These data were included in Figure 1 and in the text, section results, first §. Such a pattern of expression is common (in the cortex) for genes with known mitotic functions such as *Cyclins*, *Survivin*, and *Aurora B* (<http://www.genepaint.org/Frameset.html>), and <http://www.eurexpress.org/ee/intro.html>. See for instance, the expression pattern of Survivin pasted below). In fact, transcription pauses during the mitotic process and proteins accumulate before its onset. In line with this, cell-cycle dependent fluctuations of Diaph3 protein levels were reported (DeWard and Alberts, 2009).

In situ hybridization of a survivin/Birc5 probe on E14.5 sagittal section (extracted from Genepaint database). Note the higher expression in the outermost ventricular zone and the similarity with the Diaph3 pattern (Figure 1b,d)

It appears to be expressed at least as strongly outside the nervous system, please comment on this and the implications for non autonomous effects.

We agree with the reviewer that we cannot exclude a role for Diaph3 in other cell types. We considered the hypothesis that hypoxia can induce cortical progenitor cell death, as a result of the anemia of Diaph3 mutant embryos (Watanabe et al., 2013, and the present study), and we dedicated a substantial effort to test this possibility. We performed a specific staining using pyroninidazole hydrochloride but did not detect any sign of hypoxia in the cortical tissue at E10.5 when apoptosis peaks (Fig. S3). We believe that the phenotype we report here is autonomous to cortical progenitors as it is preserved when the gene is specifically excised in Emx1-Cre expressing cells (apoptotic cells were present in the E10.5 cortex of Diaph3 conditional ko as well, while no cell death was detected in embryos lacking either Cre recombinase or floxed allele). Emx1-Cre is expressed in dorsal telencephalon by neural progenitors which give rise to cortical excitatory neurons, but not meningeal or blood cells strongly suggesting that the cortical defect is neural progenitor cells' autonomous.

Diap3 is then inactivated and this has striking and powerful effects. Numbers of progenitors of different types and neurons were counted in different cortical regions, but these are in fact densities per unspecified field, which is meaningless since we don't know what a field is. It looks like the thickness of the cortex is reduced but density of cells is roughly normal. You need to give a better description of what the changes are in terms that are defined by standard scientific values such as microns, mm, etc.

The term "field" was specified in the "Methods" section. Nevertheless, we have revised this and expressed values as cell densities (number of cells per $100 \mu\text{m}^2$) or number of cell per $100 \mu\text{m}$ of the VZ margin.

This journal wants you to define the meaning of your error bars in the legend and you don't. You do say it in Methods. The tests used seem appropriate to me.

Thank you. The meaning of error bars has been defined in the legends

Page 8, line 4-5, I think there's a typo in the first genotype given, isn't this a control?

Thank you. The typo has been corrected in the revised manuscript.

The attempts to show the effect of deletion is autonomous are ingenious and fairly convincing, although they don't rule everything out. They do show the effects are internal to the cortex and the overall hypothesis advanced to explain the findings is very reasonable.

Overall, I think this will be of great interest to a broad range of readers and certainly it is very innovative. If the points above can be addressed thoroughly it will be an even better paper, in my opinion. It will influence thinking in this area.

The manuscript is clear and generally well written.

We thank the reviewer for acknowledging that the work is "of great interest to a broad range of readers and very innovative". We have addressed all his suggestions which helped to improve the

paper.

Reviewer #2 (Remarks to the Author):

Comments on "Lack of Diap3 relaxes the spindle checkpoint causing the loss of neural progenitors and microcephaly", by Damiani et al.

Damiani and colleagues propose that Diap3, one of 3 Diaphanous isoforms, is a key member of the regulatory pathway that decides whether neural progenitors in the developing mouse cortex should exit mitosis. For this, they generated a conditional knock-out mouse, in which they show severe and broad developmental defects and death before birth. In the cortex, they show significant cell death in neural progenitors, which they associate convincingly to severe defects in chromosome segregation. Establishing a strong causal link would, however, need some more data. The potentially most interesting point of the manuscript is the link to the SAC protein BubR1, which seems impaired from enforcing a normal spindle checkpoint in the absence of Diap3. The data for the association with CPC proteins, on the other hand, is weak. A revised manuscript could be very interesting, but also more suited to a specialized journal.

We thank the reviewer for the overall positive evaluation. We fully agree with her/his statement as regard to the link with BubR1. In the revised manuscript, we have further strengthened that link by rescue experiments and by extension to human cells (See responses to points 1-2 of reviewer #3). The proposed link with CPC proteins is indeed less strong. Yet, the observations that Diaph3 co-IP with CPC components and that Diaph3 inactivation impacts their localization are, we think, significant and of interest to our colleagues in the field. Since we do not provide further causal relationship between Diaph3 and the CPC, we have been careful in the revised manuscript to present data in a factual manner and to discuss with appropriate caution in the Discussion.

Main points:

1- This manuscript focuses on defects in the developing cortex, yet others have shown, and the authors confirm, that Diap proteins are also broadly required elsewhere, with some functional and mechanistic overlap with what the authors observe in neural progenitors. In this light, a more specialized journal seems a better fit for these more specific data.

We believe that Diaph3 is an important regulator of cell division not only in neural progenitors, but also in other fast cycling cells. In addition, the aneuploidy defect we report here and the link with the spindle assembly checkpoint protein BubR1, in mice and humans, extends the scope of the manuscript to the fields of cancer and aging. In humans, DIAPH3 is an important therapeutic target for cancer research because its downregulation i) is critical for tumor cells to acquire the amoeboid and infiltrating phenotype (Hager et al., 2013), and ii) renders cells more susceptible to poisons of microtubules such as taxanes used as chemotherapy agents (Morley et al., 2015). For all these reasons, we think that our results are of interest to a broad audience of the scientific community.

2- The full Western blots, including e.g. markers and loading controls, must be included in all figures, as key conclusions suggested by the authors rely heavily on them.

The full western blots with markers and loading controls were included in the figures and in supplementary Figure 11, which contains uncropped scans of original blots.

3- Even though a direct causal link between aneuploidy and cell death for Diap3 ko is implied, likely, and would be a key part of the study, the data for this should be more robust. One possibility would be to quantify what is shown in Fig. 4h, mitotic cells with/without segregation errors and positive/negative for cell death markers.

To quantify the ratio of apoptotic cells with nuclear segregation errors, we scrutinized 211 dividing (phospho-Vimentin⁺) cells from 3 ko embryos. One third (71) were apoptotic (activated Caspase-3 or TUNEL), and 70 had chromosomal segregation errors. On the other hand, out of the 140 remaining cells (negative for apoptosis), only 9 (6.43%) displayed mitotic errors. Hence, there is a strong relationship between apoptosis and mitotic segregation errors in Diaph3 ko cortical progenitor cells. The data have been added in the revised text

4- The validity of the data in Fig. 6 depends fully on the GFP-Diap3 fusion protein, yet validation data for this construct is completely missing. Likewise, it is unclear if overexpression per se, and overexpression differences between cells could be affecting these results (also those in Sup. Fig. 6). Therefore, little can be robustly claimed from these data about the localization and associated functionality of Diap3.

The construct used to express GFP-Diaph3 fusion protein has been generated in early 2000s. It has been extensively used and functionally validated as it was capable to:

- Rescue the activation of Serum Response Factor (SRF) upon injection of blocking antibodies (Tominaga et al., 2000) in NIH-3T3 cells.
- Rescue the cytokinesis blockade in HeLa cells induced by Diaph1 downregulation (Tominaga et al., 2000).
- Induce formation of filopodia in combination with activated Rac and Cdc42.

The GFP-Diaph3 fusion protein localizes at the tips of filopodia (Peng et al., 2003). Furthermore, using an analogous fusion protein (YFP-Diaph3), interactions of Diaph3 with Cdc42 at the MTOC (Peng et al., 2003) and with RhoB in endosomes were preserved (Wallar et al., 2007), strongly suggesting that the fusion with fluorescent proteins does not impair the localization or the function of Diaph3.

We fully agree that overexpression could in some cases alter the subcellular localization of proteins. However, we found that in cortical progenitors the localization of GFP-Diaph3 is identical/similar between cells despite differences in levels (e.g. interphasic cells in new Supplementary Fig. 10 display

different levels, but the same distribution). Genuine differences are observed between interphasic and mitotic cells.

5- Even if the construct and expression levels were found to be functional, I see no convincing "colocalization" of GFP-Diap3 with CPC components in Fig. 6 a-b. If anything, these images argue for a simple exclusion of GFP-Diap3 from the chromatin volume. A proper colocalization analysis, quantifying signals in higher resolution stacks and showing single channels would be necessary. See also point 2.

We thank the reviewer for her/his remark. One image illustrating colocalization of GFP-Diaph3 with CPC components was from an anaphase/telophase cell. At this stage, sister chromatids have been separated and Diaph3/CPC proteins localize at the central spindle (midzone) and this could be misleading in terms of association with the chromatin. In the revised manuscript, additional and high quality images from pre-anaphase cells were included. Different single channels (namely GFP-Diaph3, CPC proteins, phospho-Vimentin and DAPI) were separately shown, together with two merge images (GFP-Diaph3 and CPC proteins, phospho-Vimentin and DAPI). Colocalization analysis was performed using F10-ASW 4.0 Viewer software from Olympus. Histograms depicting fluorescence levels for the green (GFP-Diaph3) and red (CPC proteins, either Survivin or Aurora B) channels in the two different axis were generated depicting region of interests (ROIs) around the contour of the cells. Pearson correlation coefficient (PCC, or r) has also been calculated. This coefficient was always positive, strongly suggesting colocalization of GFP-Diaph3 with CPC proteins. The significance of these data was also checked according to the "Costes" method (Costes et al., 2004) with the aid of JaCOP colocalization ImageJ plugin.

Minor points:

6- The quality of the images provided should be higher, although I realize this could be due to the editorial processing of the original pictures.

Unfortunately, the procedure for the first submission of the paper required low definition figures. High resolution images have been provided for the revised version.

7- It is puzzling that the difference in Fig. 5e is deemed statistically significant, but that in Fig. 5g is not. A revision of the samples, independent sample sizes and/or statistics would seem necessary.

We thank the reviewer for her/his insightful comment. In the previous version, the differences were statistically significant or not according to the sampling of the data and the type of statistical test. We preferred to be cautious and present them as a trend not statistically significant. We have now

increased the sample size to 6 controls and 5 ko embryos (instead of 3 for each genotype) and revised the statistics (new figure 5g)

Reviewer #3 (Remarks to the Author):

Comments to the author

In this paper, the authors investigate the role of the protein Diap3 in mouse development. Diap3 is known to regulate cytokinesis, but instead they found that Diap3 is a regulator of chromosome segregation in mouse cortical progenitor cells. Loss of function of Diap3 led to massive apoptosis of neural progenitors during embryogenesis and microcephaly. The authors provide evidence for the interaction of Diap3 and BubR1, a regulator of the spindle assembly. They suggest a new molecular model highlighting the protective role of Diap3 against mitotic error induced apoptosis. The results are original and describe new functions of the Diaphanous formins, supporting Diap3 implication in primary microcephaly.

The authors have a longstanding experience in neural development, neuronal migration, cell polarity and cell division. The data and the manuscript are well presented, and the results provide proof consistent with the author's hypothesis: lack of Diap3 leads to loosening of the spindle checkpoint, enabling aneuploidy and consequent apoptosis in cortical progenitors; however, the manuscript could be improved after addressing the following major and minor changes:

We thank the reviewer for her/his positive assessment and the excellent summary.

MAJOR Changes:

1) The role of Diap3 in mouse cortical neurons is supported by the phenotypic data in the mutants, and suggests an important role in primary microcephaly. Relevant to the mutant phenotype is the interaction between Diap3 and BubR1. In this study, the authors show that in Diap3 mice, levels of BubR1 are reduced by 50%. In fact, BubR1 ko mice display some phenotypic features found in the Diap3 mice, such as progressive aneuploidy and dwarfism (Baker et al., 2004). However, the authors do not provide a rescue experiment that would point out if the aneuploidy and the small body size (including microcephaly) are a primary effect due to Diap3 absence or if they are due to the consequent reduction in BubR1 levels. The rescue experiment could be performed with *in utero* electroporation (E13.5) of BubR1 cDNA, or by transfection of BubR1 into Diap3 ko-derived neurospheres, for example.

Given that the apoptosis peaks at E10.5 and decreases as development progresses, *in utero* electroporation is technically extremely difficult and could not be performed. Therefore, we used the second alternative suggested by the Reviewer. We overexpressed BubR1 in Diap3 ko neurospheres and analyzed their karyotype. GFP and BubR1-IRES-GFP cassettes were subcloned into a pHIV

lentiviral vector. Purified lentiviruses were used to infect Diaph3-deficient progenitors. Fluorescent progenitors were cultured as neurospheres, expanded, and subjected to karyotyping. The number of aneuploid metaphases was 60.5% in GFP infected neurospheres. This percentage dropped to 33,9% in the BubR1-IRES-GFP neurospheres, indicating that BubR1 overexpression at least partially rescued the Diaph3 phenotype. The data were included in the text and in new supplementary figure 7.

The interaction among Diap3 and BubR1 was not known at the time of the BubR1 ko mouse characterization. Thus it would be helpful to test whether Diap3 levels are lower in BubR1 ko mouse cortical neuroprogenitors as well. If Diap3 levels are lower in BubR1 ko mice compared to WT, rescue experiments transfecting Diap3 into BubR1 ko mice or neurospheres should be performed. The results would clarify if the phenotype is depending solely on individual gene loss or if it is due to a combined effect of the loss of both genes.

The suggestion to test Diaph3 levels in BubR1 ko mice is indeed very interesting. However, we do not have these mice in the laboratory, and importing/deriving them (assuming that they are commercially available or that their “owners” accept to share them) would take several months.

To test whether BubR1 is necessary to stabilize Diaph3, we treated wild-type neurospheres with a pool of 4 BubR1 siRNAs (Dharmacon) and probed the level of Diaph3 by western blotting. We didn't see a decrease in Diaph3 despite the reduction of BubR1 (Supplementary Figure 8a, and text).

To check whether overexpression of Diaph3 could rescue the BubR1 induced aneuploidy, we derived MEFs from a litter of transgenic mice overexpressing Diaph3. We downregulated BubR1 repetitively for 3 passages in Diaph3 overexpressing and control MEFs and analyzed their karyotype. Knockdown of BubR1 led to a massive loss of MEFs in both genotypes. In addition, the percentage of aneuploid metaphases was 42% in control MEFs and 44,73% in Diaph3 transgenic MEFs, demonstrating that the overexpression of Diaph3 is not able to rescue/prevent the BubR1 induced aneuploidy (Supplementary Figure 8b, c, and text)

These findings are in line with published data and with our proposed model. We believe that Diaph3 acts upstream of or in parallel to BubR1 to modulate its function. BubR1 is a final effector of the checkpoint. It is a member of the mitotic checkpoint complex that binds to and inhibits the Anaphase Promoting Complex/Cyclosome (APC/C), a protein complex that functions as a poly-ubiquitin ligase for proteins such as CyclinB and Securin which inhibit the metaphase-anaphase transition and mitosis exit. BubR1 ablation generates abnormalities even at the blastocyst stage, with no embryos surviving beyond the stage of E8.5 (Wang et al., 2004), and its haploinsufficiency leads to increased susceptibility to spontaneous and induced tumorigenesis in mice (Daou et al., 2004) or mosaic variegated aneuploidy due to premature chromatid separation in humans (Matsuura et al., 2006). By contrast, 100% Diaph3 mutants survive to the stage of E10.5 and no phenotype was observed in heterozygous mice, suggesting that it is unlikely that Diaph3 acts downstream of BubR1.

2) Overexpression of DIAPH3 in humans has similar effects of Diap3 overexpression in mouse, as cited by the authors (Schoen et al., 2010). However, there are no known cases of DIAPH3 downregulation in humans, probably due to its high lethality (as shown here in the mouse model). It would be important to test if DIAPH3 depletion in human cell lines leads to progressive aneuploidy and consequent apoptosis. There are modified HEK293 cell lines available with stable knock down of

DIAPH3 that would be suitable for this purpose (Panoply[®] Human DIAPH3 Knockdown Stable Cell Line), especially due to the fact that HEK293 cells are usually hypotriploid. A reduction of chromosome number in these cells upon knockdown of DIAPH3, as well as increased apoptosis would strongly support an unidentified role for the Diap3 human homolog in miscarriages and rare cases of primary microcephaly.

We contacted the company suggested by the reviewer, but the proposed delivery time was far too long for the purpose of this revision. Thus, we knocked down DIAPH3 in HEK293T cells using Accell siRNAs (Dharmacon). This is a chemically modified siRNAs which have the capability to enter cells without transfection, allowing multiple treatments and hence long term knockdown. We treated cells for 16 days (during four passages). DIAPH3 downregulation was verified by Western blot in two different passages, and ranged from 60 % (at passage 2) to 75% (at passage 3). Interestingly, we found a concomitant reduction of BUBR1 levels, indicating that the Diaph3-BubR1 axis is preserved in human cells (Supplementary Figure 9a). Finally, we performed the karyotype analysis in control and DIAPH3 downregulated cells (Supplementary Figure 9b). In cells treated with control siRNAs, 65% of metaphases have a chromosome number ranging from 50 to 70, 15% had between 21 and 49 chromosomes, and 20% had less than 20 chromosomes. In DIAPH3-siRNA treated cells, we observed a profound decrease in the percentage of hypotriploid HEK293T cells (27% versus 65% for control had 50-70 chromosomes, n=83), along with an increase of the ratio of cells with less than 20 chromosomes (49% versus 19%). Even though these results should be considered with care given the well documented genomic instability of HEK cells (Stepanenko and Dmitrenko, 2015), they corroborate the finding obtained in *Diaph3 ko* neurospheres, and reinforce the Diaph3-BubR1 axis.

MINOR Changes:

1) Although it is clear that *Diap3 ko* mice are smaller than their littermates (Fig 1d-e), the authors do not provide their body weight and the relative head size. It is important to provide this data to discriminate between general dwarfism and primary microcephaly, and to discriminate among the overlapping phenotypic features in the *BubR1 ko* mouse.

Unfortunately, we did not make those measures at the time the five animals were produced (early when the line was established) and were not able to produce additional *ko* mice after one year or so of intercrosses probably because of inbreeding and loss of heterozygosity. We have instead generated *Diaph3;Emx1 cko* mice in which *Diaph3* is inactivated specifically in the cortex. Like the constitutive *ko*, these *cko* animals exhibit a reduction of the cortical size already at E15,5 as well as in the adult (New Supplementary Figure 5b–d), suggesting a “primary” microcephaly. We have also revised the text and restricted ourselves to terms as cortical reduction/hypoplasia to avoid the controversy related to terminology.

2) Page 8, line 4: there is a typo in the genotype of the mouse. In the sentence "Whereas no apoptosis was seen in *Diap3f/f;Emx1-Cre* or *Diap3f/f* cortices, *Diap3f/f;Emx1-Cre* embryos exhibited

many apoptotic cells, demonstrating that cell death is caused by the lack of Diap3 in cortical progenitors rather than in blood cells (Supplementary Fig. 4) should be written as "Diap3f/+;Emx1-Cre", based on Supplementary Fig.4.

Thank you. The typo has been corrected.

3) Discussion paragraph, page 12, line 4: more emphasis should be given to the sentence "These cells presumably die, depleting progressively the population of progenitors and leading to cortical atrophy (we observed a marked reduction in all cortical cell types in Diap3 ko embryos at E13.5)", as it contains part of the relevant findings of the paper. Instead, the sentence should be written as: "These cells presumably die, depleting progressively the population of progenitors and leading to cortical atrophy, as shown by the marked reduction in all cortical cell types in Diap3 ko embryos at E13.5."

The sentence has been changed as suggested. Thank you

4) Figure 4, page 27: Images for immunostaining of Tbr2/TUNEL, Pax6/TUNEL, aCas3/TUNEL are only provided for Diap3 ko cortical sections, but not for wild type cortical sections. These could be added, together with additional representative images of the same immunostainings in Diap3 cortical sections, in the Supplementary material.

A new figure (new Supplementary Figure 3) has been added as requested. It contains one example for the WT and two representative examples for the ko for the three Tbr2/TUNEL, Pax6/TUNEL and aCas3/TUNEL double stainings.

5) Figure 6, page 31: Figures a and b show two examples of mitotic cells per immunostaining, however only one of the example per immunostaining is labeled. It would be preferable to label the second example as well, creating a "a1-a2/ b1-b2" figure, or labeling the markers in the second examples as well. It would be worth pointing out that these figures refer to wild type mouse electroporated in utero.

This figure was revised. Markers were added and the different panels were labelled.

6) Supplementary information, table 1: to my understanding, the "examined" column only accounts for cells analyzed that showed chromosomal number aberration, and not the total number of examined cells (which should be 142 and 263, respectively). The addition of a column to list the total number of analyzed cells would help clarify this point. I would also suggest to add the number of chromosomes for aneuploid cells in the header (i.e. 20).

Table S1 was changed as suggested. A column was added to list the total number of examined cells, the number of euploid cells, and the number of aneuploid cells for each genotype. The number of chromosomes for aneuploid cells was indicated as a subheader in the last column.

Reviewers' comments:

Reviewer #1 (Remarks to the Author):

The authors have added significant new information and the manuscript is further improved.

Two remaining points:

1. On line 107: "Expression in intermediate progenitor (Tbr2+) cells was minimal if existing, and no signal was observed in immature neurons (Fig. 1b-e)."

I agree that Diaph3 expression overlaps Pax6 expression and, had they done double-label, they would almost certainly have found many cells labeled with both. But there is considerable overlap between the domains of Tbr2 and Diaph3 expression (Fig. 1d and e) and so, without double-label information at single-cell level, the author's statement that there is minimal or no expression in intermediate progenitor cells is unsupported and cannot be made. If the authors want to retain this statement, they should provide double-label evidence based on labeling the same section with both probes.

2. The quantification in Fig. 3c is still problematic.

In the original manuscript, they measured numbers of cells per field, stating that "The term "field" refers to a 40x acquisition field obtained with the same confocal microscope, equivalent to 317.13 μm^2 " but this did not provide sufficient information. It is possible that an acquisition field contains areas of tissue and areas of no tissue. The WT fields shown in Fig. 3a,b have more tissue in them than the ko fields, but the density of cells labelled with the markers PER UNIT AREA OF TISSUE WITHIN WHICH THESE CELLS ARE CONTAINED does not look much different. Therefore, if you express the numbers of labelled cells per field of view, obviously there are less in the ko simply because the field of view contains less tissue. In the revision, the description of "field" has been removed, the numbers in the graphs have been divided by roughly 3 to express them per 100 μm^2 instead of per 317.13 μm^2 (although confusingly the graph says it's per 100 μm , which I assume is an error, but see below). But the central problem, that the method is not clearly explained or justified, has not been addressed. It looks to me that, if you sample through a strip of cortex of constant tangential width from pia to ventricle, you will count significantly fewer cells labelled with these markers. This may be more or less what the authors are trying to say, but they need to give the details of the sampling method – maybe marking an example of a sampling area on the photographs – and express their counts unambiguously. It is possible that expressing the counts per 100 μm wide strip is the best way, since the depth of cortex varies with genotype, but whatever they do they need to explain it.

Reviewer #2 (Remarks to the Author):

Comments on the revised version of "Lack of Diap3 relaxes the spindle checkpoint causing the loss of neural progenitors and microcephaly", by Damiani et al.

The authors have done a substantial effort to improve their manuscript, and have done a good job in addressing most points raised by the reviewers. Nevertheless, there are still a couple of points that need to be clarified before the manuscript can be published.

Main point

- Regarding my original point 5, the authors have now provided higher quality images and single

channels. However, the data in the manuscript still do not make a convincing case for a specific colocalization between GFP-Diaph3 and CPC components. First, Sup. Fig. 10 does not show that GFP-Diaph3 “mainly associates with condensed chromatin”. Chromatin is not even depicted, the channels are oversaturated and the magnification is far too low for such a claim. If anything, most GFP in Fig. 6 seems excluded from the space occupied by chromatin. Second, mitotic cells shown in Fig. 6 a-d appear mainly in prometaphase, not “during the whole mitotic process”. Third and most importantly, in Fig. 6, the merge of 1 and 2 (row 3, GFP-Diaph3 and CPC), most fluorescence is either exclusively green, or red, and only a small fraction is yellow, also within the centromeric and pericentromeric heterochromatin areas. This argues strongly against a general or specific colocalization. The patterns shown are more akin to minor unspecific overlaps of fractions of both signals, and aggregates of the overexpressed GFP-Diaph3 cannot be ruled out. The authors should remove this claim from the manuscript if it is to be published.

Minor points

- Regarding my original point 3, where the authors have quantified apoptotic cells with segregation errors, a supplemental graph showing these results would be a good addition to the manuscript.
- Please indicate the nature of the “Non specific bands at 70 kDa that provide a loading control” in the necessary figures, e.g. Figure 1.

Reviewer #3 (Remarks to the Author):

The authors addressed most of the changes requested by reviewers in their revision, and the new results highlight the connection of Diaph3 and the regulation of the spindle checkpoint, confirming the author’s hypothesis and improving the overall quality of the paper. Each of the reviewers suggested experiments and changes that were necessary, and nearly all of those comments were addressed, strengthening the author’s conclusions. The rescue and knockdown experiments that we suggested are shown in supplementary figures 7, 8 and 9. The results are convincing and highlight the synergic function of Diaph3 and BubR1.

However, there are still two minor changes that we believe are needed:

- 1) The weight and body size of the mice is not shown in the results (page 6). In figure 1g-i, it is clear that the Diaph3 ko/ko mice have severe developmental delay, but the head size does not appear to be smaller than expected based on the body size. Although the brain (mainly the cortex) is affected, it is also possible that these mice display generalized dwarfism with hydrocephalous, rather than true microcephaly.
- 2) Figure 1a: while in the text the following developmental dates are mentioned: E10.5, E13.5, E16.5, P0, in the figure only E10.5 and E13.5 are shown. The main text should be changed accordingly.

Response to Referees:

Reviewers' comments:

Reviewer #1 (Remarks to the Author):

The authors have added significant new information and the manuscript is further improved.

We thank the reviewer for her/his positive evaluation

Two remaining points:

1. On line 107: "Expression in intermediate progenitor (Tbr2+) cells was minimal if existing, and no signal was observed in immature neurons (Fig. 1b-e)."

I agree that Diaph3 expression overlaps Pax6 expression and, had they done double-label, they would almost certainly have found many cells labeled with both. But there is considerable overlap between the domains of Tbr2 and Diaph3 expression (Fig. 1d and e) and so, without double-label information at single-cell level, the author's statement that there is minimal or no expression in intermediate progenitor cells is unsupported and cannot be made. If the authors want to retain this statement, they should provide double-label evidence based on labeling the same section with both probes.

We fully agree with the reviewer. We have adapted the text to take into account her/his remark and removed the statement of "minimal or no expression" of Diaph3 in intermediate progenitors.

2. The quantification in Fig. 3c is still problematic.

In the original manuscript, they measured numbers of cells per field, stating that "The term "field" refers to a 40x acquisition field obtained with the same confocal microscope, equivalent to 317.13 μm^2 " but this did not provide sufficient information. It is possible that an acquisition field contains areas of tissue and areas of no tissue. The WT fields shown in Fig. 3a,b have more tissue in them than the ko fields, but the density of cells labelled with the markers PER UNIT AREA OF TISSUE WITHIN WHICH THESE CELLS ARE CONTAINED does not look much different. Therefore, if you express the numbers of labelled cells per field of view, obviously there are less in the ko simply because the field of view contains less tissue. In the revision, the description of "field" has been removed, the numbers in the graphs have been divided by roughly 3 to express them per 100 μm^2 instead of per 317.13 μm^2 (although confusingly the graph says it's per 100 μm , which I assume is an error, but see below). But the central problem, that the method is not clearly explained or justified, has not been addressed. It looks to me that, if you sample through a strip of cortex of constant tangential width from pia to ventricle, you will count significantly fewer cells labelled with these markers. This may be more or less what the authors are trying to say, but they need to give the details of the sampling method – maybe marking an example of a sampling area on the photographs - and express their counts unambiguously. It is possible that expressing the counts per 100 μm wide strip is the best way, since the depth of cortex varies with genotype, but whatever they do they need

to explain it.

We apologize for this confusion. Counts are indeed expressed as the number of cells per 100µm wide cortical stripe. An equivalent area has been boxed in Fig. 3a. We have adapted the text and figure labels & legend to explain the sampling method.

Reviewer #2 (Remarks to the Author):

Comments on the revised version of “Lack of Diap3 relaxes the spindle checkpoint causing the loss of neural progenitors and microcephaly”, by Damiani et al.

The authors have done a substantial effort to improve their manuscript, and have done a good job in addressing most points raised by the reviewers.

Thank you.

Nevertheless, there are still a couple of points that need to be clarified before the manuscript can be published.

Main point

- Regarding my original point 5, the authors have now provided higher quality images and single channels. However, the data in the manuscript still do not make a convincing case for a specific colocalization between GFP-Diaph3 and CPC components. First, Sup. Fig. 10 does not show that GFP-Diaph3 “mainly associates with condensed chromatin”. Chromatin is not even depicted, the channels are oversaturated and the magnification is far too low for such a claim. If anything, most GFP in Fig. 6 seems excluded from the space occupied by chromatin. Second, mitotic cells shown in Fig. 6 a-d appear mainly in prometaphase, not “during the whole mitotic process”. Third and most importantly, in Fig. 6, the merge of 1 and 2 (row 3, GFP-Diaph3 and CPC), most fluorescence is either exclusively green, or red, and only a small fraction is yellow, also within the centromeric and pericentromeric heterochromatin areas. This argues strongly against a general or specific colocalization. The patterns shown are more akin to minor unspecific overlaps of fractions of both signals, and aggregates of the overexpressed GFP-Diaph3 cannot be ruled out. The authors should remove this claim from the manuscript if it is to be published.

We agree with the reviewer’s comment. We have removed the claim of co-localization of Diaph3 with CPC components and the corresponding panels (Fig 6a-d) as requested. The text was rephrased as follows: “The signal was perinuclear and extended to apical and basal processes in non-dividing - cells, and more diffusely distributed around the chromatin in dividing cells”.

Minor points

- Regarding my original point 3, where the authors have quantified apoptotic cells with segregation errors, a supplemental graph showing these results would be a good addition to the manuscript.

A graph summarizing this finding has been added in Supplementary Figure 3.

- Please indicate the nature of the “Nonspecific bands at 70 kDa that provide a loading control” in the necessary figures, e.g. Figure 1.

We have no idea about the nature of the nonspecific bands. They do not, however, reflect short forms of Diaph3 because the antibody is directed against a region that is absent in the *ko*. We have labeled them as unspecific bands.

Reviewer #3 (Remarks to the Author):

The authors addressed most of the changes requested by reviewers in their revision, and the new results highlight the connection of Diaph3 and the regulation of the spindle checkpoint, confirming the author's hypothesis and improving the overall quality of the paper. Each of the reviewers suggested experiments and changes that were necessary, and nearly all of those comments were addressed, strengthening the author's conclusions. The rescue and knockdown experiments that we suggested are shown in supplementary figures 7, 8 and 9. The results are convincing and highlight the synergic function of Diaph3 and BubR1.

Thank you.

However, there are still two minor changes that we believe are needed:

1) The weight and body size of the mice is not shown in the results (page 6). In figure 1g-i, it is clear that the Diaph3 *ko/ko* mice have severe developmental delay, but the head size does not appear to be smaller than expected based on the body size. Although the brain (mainly the cortex) is affected, it is also possible that these mice display generalized dwarfism with hydrocephalous, rather than true microcephaly.

The reviewer is right. We have removed the word "microcephaly" and rewritten the sentence as follows: "Surviving animals displayed small brain and body size, as well as facial deformities". "Microcephaly" was also removed from the title.

2) Figure 1a: while in the text the following developmental dates are mentioned: E10.5, E13.5, E16.5, P0, in the figure only E10.5 and E13.5 are shown. The main text should be changed accordingly.

Thank you. This has been changed.